# SLC1A3 contributes to L-asparaginase resistance in solid tumors

Jianhui Sun[1,2], Remco Nagel[1], Esther A Zaal[3], Alejandro Piñeiro Ugalde[1], Ruiqi Han[1,2], Natalie Proost[4], Ji-Ying Song[5], Abhijeet Pataskar[1], Artur Burylo[6], Haigen Fu[7], Gerrit J Poelarends[7], Marieke van de Ven[4], Olaf van Tellingen[6], Celia R Berkers[3,8] & Reuven Agami[1,2,*]

## Abstract

L-asparaginase (ASNase) serves as an effective drug for adolescent acute lymphoblastic leukemia. However, many clinical trials indicated severe ASNase toxicity in patients with solid tumors, with resistant mechanisms not well understood. Here, we took a functional genetic approach and identified SLC1A3 as a novel contributor to ASNase resistance in cancer cells. In combination with ASNase, SLC1A3 inhibition caused cell cycle arrest or apoptosis, and myriads of metabolic vulnerabilities in tricarboxylic acid (TCA) cycle, urea cycle, nucleotides biosynthesis, energy production, redox homeostasis, and lipid biosynthesis. SLC1A3 is an aspartate and glutamate transporter, mainly expressed in brain tissues, but high expression levels were also observed in some tumor types. Here, we demonstrate that ASNase stimulates aspartate and glutamate consumptions, and their refilling through SLC1A3 promotes cancer cell proliferation. Lastly, *in vivo* experiments indicated that SLC1A3 expression promoted tumor development and metastasis while negating the suppressive effects of ASNase by fueling aspartate, glutamate, and glutamine metabolisms despite of asparagine shortage. Altogether, our findings identify a novel role for SLC1A3 in ASNase resistance and suggest that restrictive aspartate and glutamate uptake might improve ASNase efficacy with solid tumors.

**Keywords** aspartate/glutamate; genome-wide CRISPR screen; L-asparaginase; SLC1A3; solid tumors

**Subject Categories** Cancer; Metabolism

The EMBO Journal (2019) 38: e102147

## Introduction

Treating cancer with amino acid deprivation schemes has achieved limited clinical success so far. Only in acute lymphoblastic leukemia (ALL), the incorporation of L-asparaginase (ASNase) has significantly increased the overall survival rates to ~90% (Broome, 1961; Müller & Boos, 1998; Pui *et al*, 2009). ALL cells are auxotrophic for asparagine, which was deaminated and depleted by the enzyme ASNase, resulting in cell cycle arrest and apoptosis in ALL cells without affecting normal tissues (Kidd, 1953; Broome, 1961; Ueno *et al*, 1997; Pui *et al*, 2009). Notably, ASNase has a dual asparagine and glutamine deaminase activity; however, its glutaminase activity was not required for anticancer effect in asparagine synthetase (ASNS)-negative cancer cells (Chan *et al*, 2014). The therapeutic progress of ASNase in ALL had greatly encouraged its further application for solid tumors. However, many clinical trials reported intolerable toxicity in patients (Haskell *et al*, 1969; Hays *et al*, 2013). ASNS expression has been proposed as a marker for clinical prediction of ASNase resistance (Scherf *et al*, 2000); however, treatment of ALL with ASNase is still effective even though ASNS is expressed (Stams, 2003; Krall *et al*, 2016; Vander Heiden & DeBerardinis, 2017). Interestingly, aspartate metabolism was also predicted to contribute to ASNase sensitivity according to a previous study (Chen *et al*, 2011). Overall, with the exception of ASNS, little is known about the specific resistant mechanisms to ASNase, which has hindered the attempts to broaden ASNase's benefits to patients with solid tumors (Kidd, 1953; Haskell *et al*, 1969; Hays *et al*, 2013; Vander Heiden & DeBerardinis, 2017).

Our previous work has found that ASNase treatment of PC3, a prostate cancer cell line, triggered asparagine shortage accompanied by increased asparagine production through upregulation of ASNS, as indicated by ribosomal and transcriptional profiling (Loayza-Puch *et al*, 2016). This pinpointed a feedback loop under asparagine depleted conditions. Yet, PC3 cells remained proliferative despite of

1  Division of Oncogenomics, Oncode Institute, The Netherlands Cancer Institute, Amsterdam, The Netherlands
2  Department of Genetics, Erasmus University Medical Center, Rotterdam, The Netherlands
3  Biomolecular Mass Spectrometry and Proteomics, Bijvoet Center for Biomolecular Research, Utrecht University, Utrecht, The Netherlands
4  Preclinical Intervention Unit and Pharmacology Unit of the Mouse Clinic for Cancer and Ageing (MCCA), The Netherlands Cancer Institute, Amsterdam, The Netherlands
5  Division of Experimental Animal Pathology, The Netherlands Cancer Institute, Amsterdam, The Netherlands
6  Division of Pharmacology, The Netherlands Cancer Institute, Amsterdam, The Netherlands
7  Department of Chemical and Pharmaceutical Biology, University of Groningen, Groningen, The Netherlands
8  Department of Biochemistry and Cell Biology, Faculty of Veterinary Medicine, Utrecht University, Utrecht, The Netherlands
   *Corresponding author. Tel: +31 205122079; E-mail: r.agami@nki.nl

asparagine shortage, suggesting the involvement of other mechanisms responsible for ASNase resistance as upregulated ASNS was not sufficient for asparagine replenishment. Therefore, we used a functional genetic screen in PC3 cells to explore potential vulnerabilities in solid cancer cells to ASNase treatment. We identified SLC1A3, an aspartate/glutamate transporter, as a novel contributor to ASNase resistance, as well as tumor initiation and progression in a mice model for breast cancer metastasis.

# Results

### A genome-wide CRISPR-Cas9 screen identifies SLC1A3 as a novel contributor to ASNase resistance in PC3 cells

To determine the optimal ASNase concentration required for performing a genome-wide functional screen, we tested a series of ASNase concentrations in PC3 cells. Figure 1A shows that ASNase at a concentration of 0.3~0.5 U/ml moderately inhibited cell proliferation. As this dosage is within the range used for asparagine depletion in some ALL patients according to previous research (Riccardi *et al*, 1981; Avramis & Panosyan, 2005), we performed the screen and *in vitro* validation under this condition. Due to its essential role in asparagine synthesis, ASNS gene was used as a positive control for the screen. As expected, CRISPR-Cas9 knockout (KO) of ASNS sensitized PC3 cells to ASNase treatment but did not affect cell proliferation under mock treatment (Fig 1B).

Next, we transduced a genome-wide CRISPR-Cas9 library, consisting of 76,441 single-guide RNAs (sgRNAs) targeting 19,114 genes, into PC3 cells, which were further divided into mock- and ASNase-treated conditions (Fig 1C). Following 20 days of culturing, cells were harvested and subjected to deep sequencing of integrated sgRNAs and MAGeCK bioinformatics analysis of individual sgRNA abundance. Intriguingly, in addition to the expected ASNS gene, this analysis proposed four additional genes (FDR < 0.003, Fig 1D), whose loss of function may impair PC3 cell proliferation following ASNase treatment. Follow-up validations using individual CRISPR vector transductions and cell competitive growth assays successfully validated three out of the four additional hits: EIF2AK4 (GCN2, general control nonderepressible 2), SLC1A3, and SLC25A1 (Figs 1D and EV1A), highlighting the reliability of the screen. Notably, EIF2AK4 was also predictable due to its role in regulating general nutrient deprivation responses (Bunpo *et al*, 2009; Ye *et al*, 2010). The other two hits (SLC1A3 and SLC25A1) are both from the solute carrier family (SLC). SLC1A3 functions as a high-affinity aspartate and glutamate transporter, whose loss of function triggered a marked reduction in cell survival and proliferation following ASNase treatment (Figs 1E and EV1B), and SLC25A1 functions as a mitochondria citrate carrier, whose loss of function also caused inhibitory effects on cell survival and proliferation in the presence of ASNase, but to a more moderate extent when compared with that of SLC1A3 (Fig EV1A). Due to the relatively strong synergistic effect, from now on, we only focused on the role of SLC1A3 in the context of ASNase.

SLC1A3 is mainly expressed in brain tissues (Fig EV1C), critical for the termination of excitatory neurotransmission (Kanai *et al*, 2013). Recent studies have highlighted the importance of SLC1A3-mediated aspartate uptake for cancer cell proliferation under hypoxia and crosstalk between cancer cells and cancer-associated fibroblasts in the tumor niche (Alkan *et al*, 2018; Garcia-Bermudez *et al*, 2018; Sullivan *et al*, 2018; Tajan *et al*, 2018; Bertero *et al*, 2019). We also observed elevated SLC1A3 RNA levels in several tumor types from the TCGA database [especially kidney renal clear cell carcinoma (KIRC, $P = 5.5 \times 10^{-30}$), kidney renal papillary cell carcinoma (KIRP, $P = 2.1 \times 10^{-10}$), liver hepatocellular carcinoma (LIHC, $P = 3.2 \times 10^{-10}$), and stomach adenocarcinoma (STAD, $P = 6.1 \times 10^{-5}$)] (Fig EV1D).

To examine the function of SLC1A3, we tested its cellular aspartate/glutamate transporting function using a radioactive labeled amino acid uptake assay as previously described (Loayza-Puch *et al*, 2017). As predicted, SLC1A3 loss of function reduced both aspartate and glutamate uptake in PC3 cells (Fig 1F), also leading to decreased endogenous aspartate (~8-fold) and glutamate (~1.5-fold) levels (Fig 1G). Following ASNase treatment in control PC3 cells, we observed strong depletions of both asparagine and glutamine (Fig 1G), in concordance with its known dual functions. This was followed by a significant reduction in endogenous aspartate and glutamate levels (Fig 1G), indicating a stimulated demand for aspartate and glutamate. Consequently, in SLC1A3-KO PC3 cells, aspartate and glutamate levels were further depleted under ASNase treatment (~16-fold for aspartate and ~3-fold for glutamate, Fig 1G). This observation suggests that SLC1A3-mediated aspartate and glutamate import is required for the maintenance of sufficient intracellular aspartate and glutamate pools to survive ASNase treatment. Of note, the endogenous glutamine level was significantly depleted in SLC1A3-KO PC3 cells, but this had no effect on cell proliferation in the absence of ASNase (Fig 1G and E). To directly test the functions of aspartate and glutamate in the context of ASNase, we supplemented SLC1A3-KO PC3 cells with cell-permeable forms of aspartate and glutamate (esterified). Figure 1H shows that both esterified aspartate and esterified glutamate, but not esterified leucine (control), can restore SLC1A3-KO PC3 cell proliferation in the presence of ASNase. Lastly, we examined a possible role of SLC1A3 to ASNase treatment *in vivo*. We subcutaneously implanted control and SLC1A3-KO PC3 cells into Balb/c nude mice (cAnN/Rj) and examined tumor growth in the absence and presence of ASNase. Figure EV1E shows that loss of SLC1A3 in combination of ASNase treatment impeded tumor growth. Altogether, we conclude that SLC1A3 expression negates the impact of ASNase on PC3 cell survival, proliferation, and tumor growth.

### SLC1A3 mRNA levels correlate with ASNase sensitivity in different cancer cells

Because SLC1A3 transports both aspartate and glutamate (Fig 1F and G), we mainly used aspartate uptake as a functional readout for SLC1A3 in further study. We investigated the correlations between SLC1A3 mRNA level, aspartate uptake, and sensitivity to ASNase treatment in a panel of prostate and breast cancer cell lines. As predicted, we observed a general trend where relatively high SLC1A3 mRNA levels indicated high basal aspartate uptake capability (Fig 2A and B). The exceptions in our cohort were LNCaP, SUM159PT, and BT549 cells, with low SLC1A3 mRNA level but high basal aspartate uptake capacity. This can be explained by the relatively high expression of other aspartate/glutamate transporters in these cells (Fig 2C). Accordingly, SLC1A3-KO reduced aspartate uptake level only in SLC1A3-expressing cancer cells (Fig 2A and B). Interestingly, the sensitivity profiles of the tested cancer cell lines to

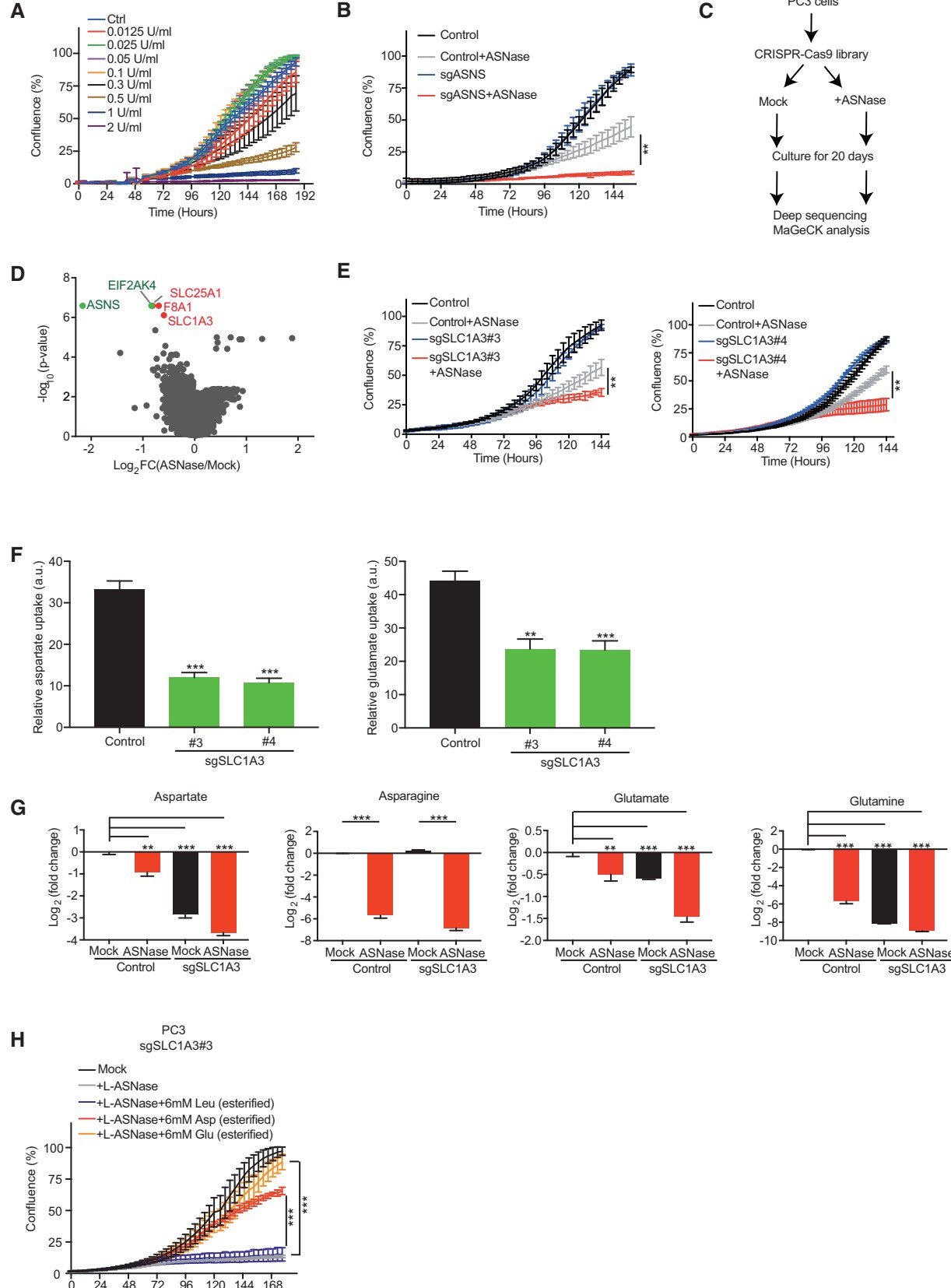

Figure 1.

◀

**Figure 1.   A genome-wide CRISPR-Cas9 screen identifies SLC1A3 as a contributor to L-asparaginase (ASNase) resistance in PC3 cells.**

A   IncuCyte cell proliferation curves of PC3 cells treated with the indicated concentrations of ASNase.

B   IncuCyte cell proliferation curves for ASNS knockout (sgASNS) and control (sgNon-targeting) PC3 cells in the absence and presence of ASNase.

C   Flow chart for a genome-wide CRISPR-Cas9 functional screen in PC3 cells.

D   Volcano plots for the MAGeCK pipeline analysis of the sgRNA abundance from the screen. Green dots indicate positive controls and red dots indicate candidates with a fold discovery rate (FDR) < 0.003.

E   IncuCyte cell proliferation curves of SLC1A3 knockout (sgSLC1A3) and control (sgNon-targeting) PC3 cells in the absence and presence of ASNase treatment. #3 and #4 represent two different sgRNAs targeting SLC1A3.

F   Radioactive labeled aspartate and glutamate uptake measurement in control (sgNon-targeting) and SLC1A3 knockout (sgSLC1A3) PC3 cells. #3 and #4 represent two different sgRNAs targeting SLC1A3. Radioactive labeled leucine uptake was used as a control. Data were normalized to the reads of control PC3 cells.

G   Endogenous levels of aspartate, asparagine, glutamate, and glutamine in control (sgNon-targeting) and SLC1A3 knockout (sgSLC1A3) PC3 cells with or without ASNase for 3 days. Median peak intensity was used for the read normalization.

H   IncuCyte cell proliferation curves of SLC1A3 knockout (sgSLC1A3#3) PC3 cells treated with ASNase and supplemented with either esterified aspartate (Asp, 6 mM) or esterified glutamate (Glu, 6 mM), and esterified leucine (Leu, 6 mM) as a control.

Data information: For IncuCyte proliferation assays, images were taken every 4 h and the cell confluence was calculated by averaging three mapped images per well. All results were calculated from three replicates and presented as mean ± SD, unless otherwise stated. The *P*-value was calculated by two-tailed unpaired *t*-test by Prism7. **$P < 0.01$, ***$P < 0.001$.

ASNase treatment were generally consistent with the impact of SLC1A3-KO on aspartate uptake, with the exception of BT549 cells (Fig 2B and D). To further confirm the correlation between aspartate/glutamate uptake capacity and ASNase sensitivity, we used SLC1A3-deficient MCF7 cells (a breast cancer cell line) and DU145 cells (a prostate cancer cell line), and established two cancer cell lines overexpressing SLC1A3: MCF7-V5-SLC1A3 and DU145-V5-SLC1A3. Figure 2E–G verified SLC1A3 ectopic expression, its subcellular localization to the plasma membrane, and its capacity to uptake up aspartate in those two cell lines. Importantly, acquired ASNase resistance was observed in both cell lines after the ectopic expression of SLC1A3 (Fig 2H). In line with the above results, the addition of cell-permeable aspartate and glutamate, but not esterified leucine, to DU145 cells, restored cell proliferation under ASNase conditions (Fig 2I). Taken together, we conclude that SLC1A3-mediated aspartate/glutamate uptake promoted ASNase resistance.

**Combination of SLC1A3 inhibition and ASNase induces metabolic vulnerabilities that impede cancer cell proliferation**

Next, we assessed chemical SLC1A3 inhibition in the context of ASNase. We mainly compared two SLC1A3 inhibitors, the selective non-substrate blocker UCPH-101 (Abrahamsen *et al*, 2013) and TFB-TBOA (Shimamoto *et al*, 2004). By aspartate uptake assay, we observed that the inhibitory activity of TFB-TBOA was far more potent than that of UCPH-101, even reaching a nanomolar level (Fig EV2A). Therefore, we used TFB-TBOA for further experiments. Notably, while TFB-TBOA and ASNase, respectively, had either no or mild effect on PC3 cell proliferation, their combinational treatment effectively hindered cell proliferation and cell cycle progression (Figs 3A and B, and EV2B–D). In addition, TFB-TBOA completely restored the adverse effect of ASNase in DU145-V5-SLC1A3 cells but had no influence on DU145 wild-type cells (Fig 3A and C). Interestingly, while the combinational treatment impaired cell cycle progression in PC3 cells, it caused apoptosis in DU145-V5-SLC1A3 cells (Figs 3C and EV2E).

Next, we investigated the effects of combined ASNase and SLC1A3 inhibition on intracellular amino acids and key metabolites levels by liquid-chromatography mass spectrometry (LC-MS). In concordance with SLC1A3-KO, SLC1A3 inhibition by TFB-TBOA promoted further depletions of aspartate and glutamate pools in PC3

cells in the presence of ASNase (Fig 3D). However, in contrast to SLC1A3-KO, TFB-TBOA did not perturb aspartate, glutamate, and glutamine levels, probably due to the short drug exposure time compared with the genetic manipulation. Notably, combined ASNase and SLC1A3 inhibition induced a marked reduction in argininosuccinate from the urea cycle (Fig 3E). This effect can be reasoned by the lack of aspartate availability as a substrate for argininosuccinate synthesis (Rabinovich *et al*, 2015). Moreover, nucleotide synthesis and tricarboxylic acid (TCA) cycle replenishments were also impaired (Fig 3E), probably due to the deficit of aspartate as previously described (Ahn & Metallo, 2015; Rabinovich *et al*, 2015). We also observed that combinational treatment disturbed the $NAD^+$/NADH homeostasis, an important indicator for cellular energy assessment and redox status (Fig 3E). And strong lactate depletion was detected, at least partly due to the depletion of NADH (Fig 3E). Moreover, levels of carnitine metabolites (important transporters for lipid metabolism) were also perturbed under combinational conditions (Fig 3E). Above all, in SLC1A3-expressed PC3 cells, ASNase and TFB-TBOA impact metabolites involved in the urea cycle, nucleotides synthesis, energy production by TCA cycle and glycolysis, as well as redox homeostasis and lipid metabolism. These metabolic alterations further explain why PC3 cells resist ASNase treatment but become vulnerable once SLC1A3 is either genetically depleted or chemically blocked.

We also probed the key metabolites in DU145 and DU145-V5-SLC1A3 cells. In DU145 cells (lacking SLC1A3 expression), ASNase alone was sufficient to induce a similar metabolic profile as obtained in PC3 treated with ASNase and SLC1A3 inhibition (Fig EV2F). Ectopic expression of SLC1A3 negated these adverse effects, and accordingly, the addition of TFB-TBOA restored those perturbations in DU145-V5-SLC1A3 cells (Fig EV2F).

Then, we inquired whether ASNase treatment promotes a special usage of cellular aspartate/glutamate. For that purpose, we conducted metabolic flux studies using $[^{13}C_4,^{15}N]$ L-aspartate and $[^{13}C_5,^{15}N]$ L-glutamate in DU145-V5-SLC1A3 cells. Notably, as observed before (Sullivan *et al*, 2018), exogenous labeled aspartate was barely incorporated to the intracellular asparagine pool (Fig EV3A). Instead, both labeled aspartate and glutamate were actively used to replenish downstream metabolisms, such as TCA cycle, urea cycle, and nucleotide synthesis. However, following ASNase treatment, the relative profiles of labeled metabolites

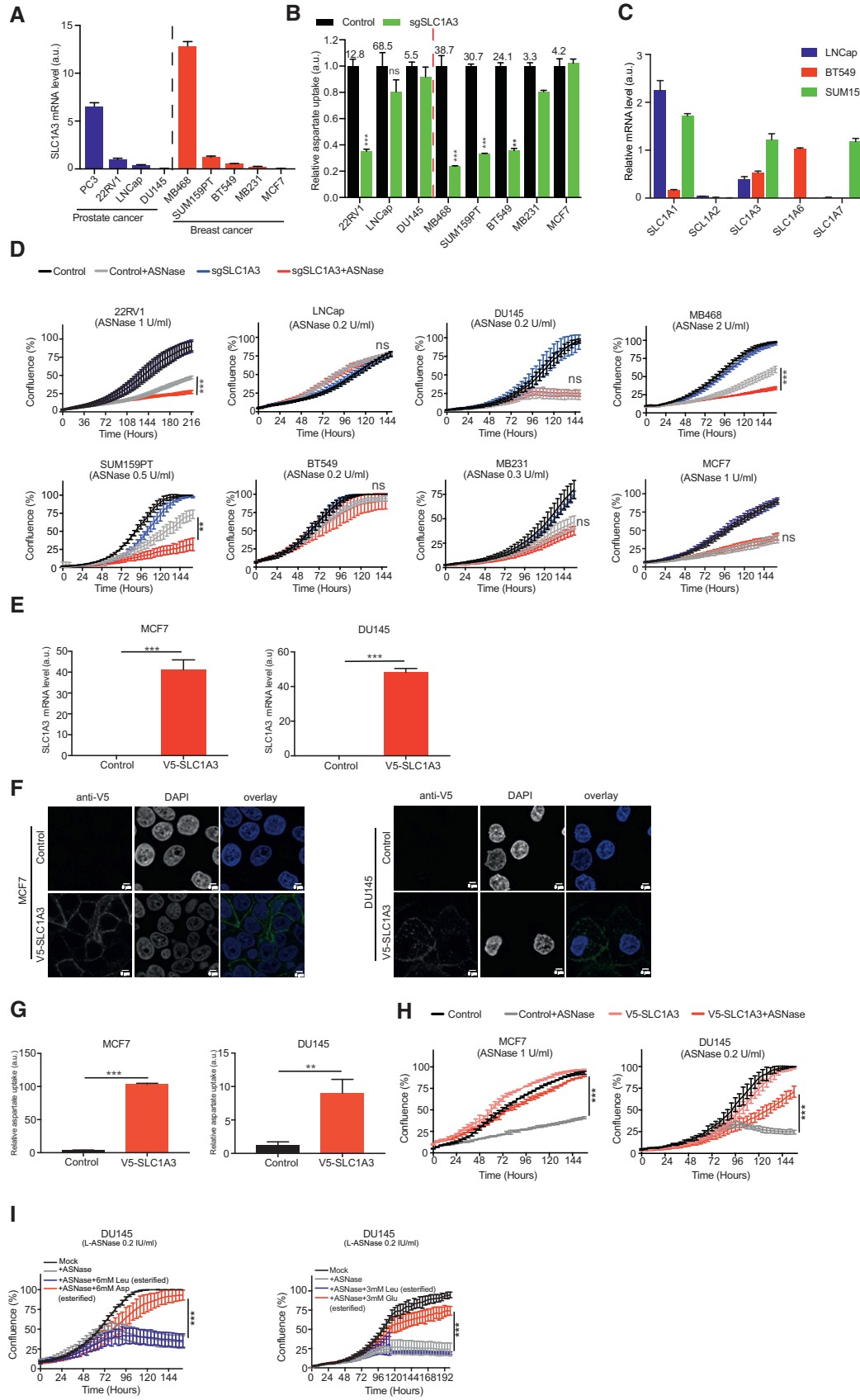

**Figure 2.**

**Figure 2.  SLC1A3 expression is linked to ASNase resistance in different cancer cells.**

A   RT–qPCR analysis was used to determine the relative SLC1A3 mRNA expression (to GAPDH) in different prostate and breast cancer cell lines, as indicated.
B   The same cell lines (as in panel A) were transduced with either control (sgNon-targeting) or sgSLC1A3. Aspartate uptake levels were determined and compared between control and SLC1A3 KO in these cell lines. Leucine uptake level was used for normalization. The numbers above the control column denote the basal aspartate uptake capacity.
C   RT–qPCR was used to determine the relative mRNA levels (to GAPDH) of aspartate/glutamate transporter genes (SLC1A1, SLC1A2, SLC1A3, SLC1A6, and SLC1A7) in LNCaP, BT549, and SUM159PT cells.
D   The same batch of cancer cells (as in panel B) was subjected to IncuCyte cell proliferation assays in the absence or presence of ASNase at indicated concentrations. "ns" indicates no significant difference.
E   MCF7 and DU145 cells were transduced with either lentiviral empty vector (control) or lentiviral vector containing a V5-tagged SLC1A3 coding sequence (V5-SLC1A3). Relative SLC1A3 mRNA levels (to GAPDH) were determined by RT–qPCR.
F   Immunofluorescence staining of the V5-tagged SLC1A3 in MCF7 and DU145 cells using anti-V5 antibody. Green staining indicates the plasma membrane localization of V5-SLC1A3 and blue DAPI staining marks the nuclei. Scale bar stands for 5 μm.
G   Relative aspartate uptake levels in control and V5-SLC1A3-expressed MCF7 and DU145 cells. Leucine uptake level was used for normalization.
H   Control and V5-SLC1A3-expressed MCF7 and DU145 cells were subjected to IncuCyte cell proliferation assays with or without ASNase at indicated concentrations.
I   DU145 cells were supplemented with cell-permeable aspartate (Asp, 6 mM, esterified) or glutamate (Glu, 3 mM, esterified) following ASNase treatment, with esterified leucine (Leu, 6 or 3 mM) as control.

Data information: Results were calculated based on three replicates (except for SUM159 and BT549 in B, $n = 2$) and presented as mean ± SD. The P-value was calculated by two-tailed unpaired t-test in Prism7. **$P < 0.01$, ***$P < 0.001$. a.u. indicates arbitrary unit.

remained generally similar to that of mock-treated cells, except for increased incorporation into glutamine from labeled glutamate (Fig EV3B and C). Thus, ASNase treatment did not induce significant perturbations in the general metabolic usage of aspartate and glutamate in cancer cells.

### Gene expression analysis indicates the novel role of SLC1A3 in ASNase resistance

To interrogate the influence on differential gene expression profiles by SLC1A3 and ASNase, we performed transcriptome analysis in three cancer cell lines: PC3 (endogenous expression of SLC1A3), DU145 wild type (SLC1A3 negative), and DU145-V5-SLC1A3 (ectopic expression of SLC1A3). Consistent with the compromised cell cycle progression in PC3 cells (Figs 3B and EV2B–D), genes related to cell cycle progression were inhibited following ASNase and TFB-TBOA combinational treatment (Fig 4A). ASNase-treated DU145 cells presented upregulated apoptotic signatures, corresponding to the apoptosis phenotype observed in these cells following ASNase treatment (Figs 4B and 3C). Intriguingly, the introduction of SLC1A3 to DU145 cells prevented the emergence of the apoptosis signature, which was restored by the addition of TFB-TBOA (Figs 4C and 3B). The molecular pathways related to lipid metabolism, for example, the biosynthesis of cholesterol, steroid and mevalonate, and the gene expression related to sterol regulatory element-binding protein (SREBP), have been strongly impaired in all three cancer cell lines when SLC1A3 was either chemically blocked or intrinsically absent following ASNase treatment (Fig 4A–C).

More specifically, we observed increased vascular endothelial growth factor A (VEGFA) mRNA levels after combinational treatment in PC3 cells (Fig EV4A), in line with previous observations that suggested a negative correlation between VEGFA and aspartate level (Garcia-Bermudez et al, 2018). Moreover, the decreased mRNA level of lactate dehydrogenase A (LDHA) can explain the depletion in lactate level measured by metabolites profiling (Fig 3E). Though ASNS mRNA level was strongly upregulated by ASNase and SLC1A3 inhibition in DU145-V5-SLC1A3 cells, still, cell death was induced (Fig EV4C). This indicates that elevated asparagine synthesis by ASNS could be insufficient to convey ASNase resistance,

which might also be determined by aspartate/glutamate bioavailability.

Altogether, we conclude that the transcriptomic changes (Fig 4A–C) are in concordance with metabolomic perturbations (Figs 3E and EV2F) and cellular outcomes (Fig 3A–C), indicating a novel role of SLC1A3 in cancer cell survival following ASNase treatment.

### SLC1A3 expression promotes tumor progression and ASNase resistance in a mouse model for breast cancer metastasis

Next, we set up experiments to examine the role of SLC1A3 in tumor response to ASNase treatment *in vivo*. As a first step, we interrogated the impact of ASNase treatment on asparagine and glutamine levels in mice with human breast cancer xenografts. We orthotopically injected human breast cancer cells (SUM159PT) to mammary fat pads of NOD-Scid IL2Rg-null (NSG) mice, allowed tumors to develop to ~250 mm³, and then systemically injected 60 U ASNase per day for 5 consecutive days. Remarkably, we detected very strong ASNase-induced depletions of asparagine not only in the blood, but also in the mammary fat pad tissues and even within the growing tumors (Fig 5A). However, unlike the glutaminase effect of ASNase *in vitro*, here we detected very modest perturbation in glutamine levels (Fig EV5A). This is probably due to the instant glutamine replenishment under *in vivo* conditions. ASNase treatment could potentially disturb tumor growing environment, at least in the perspective of asparagine.

Next, we employed 4T1, a highly malignant mouse breast cancer cell line, for the assessment of the influence of SLC1A3 on ASNase efficacy *in vivo*. This cell line does not express SLC1A3 (www.biogps.org), does not take up aspartate, and accordingly shows high sensitivity to ASNase treatment (Fig EV5B). As expected, ectopic expression of SLC1A3 (4T1-V5-SLC1A3) promoted exogenous aspartate uptake and restored 4T1 proliferation in the presence of ASNase *in vitro* (Fig EV5B). We therefore implanted 4T1 and 4T1-V5-SLC1A3 cells into the mammary fat pad of either mock- or ASNase-pretreated NSG mice and measured tumor development. Intriguingly, while the growth of tumors derived from parental 4T1 cells was impaired by ASNase at an early stage (days 9 and 12), SLC1A3-expressing tumors showed no significant differences

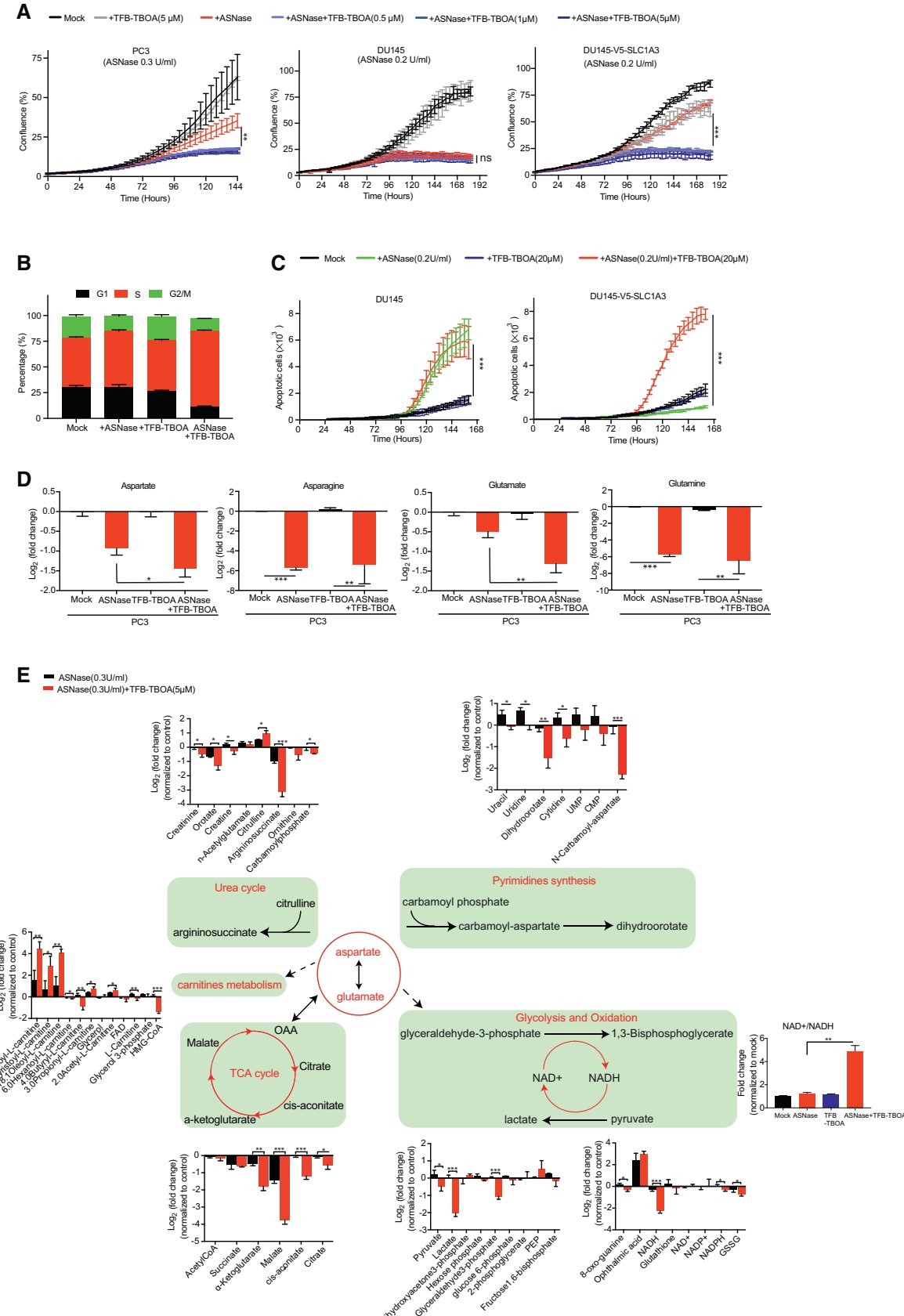

**Figure 3.**

**Figure 3. Combinational treatment of ASNase and SLC1A3 inhibition induced metabolic vulnerabilities and restrains cancer cell proliferation.**

A  PC3, DU145, and V5-SLC1A3-DU145 cells were subjected to ASNase and TFB-TBOA treatment at indicated concentrations, and cell proliferation was measured by IncuCyte assay.

B  PC3 cells were treated under indicated conditions for 9 days and subjected to BrdU assays to determine cell cycle distributions. ASNase (0.3 U/ml), TFB-TBOA (5 μM).

C  DU145 and V5-SLC1A3-DU145 cells were treated under indicated conditions with ASNase (0.2 U/ml) or TFB-TBOA (20 μM) or both, and subjected to IncuCyte analysis for apoptotic cell counts.

D  PC3 cells were treated under ASNase (0.3 U/ml), or TFB-TBOA (5 μM) conditions for 3 days and cell lysates were extracted and intracellular contents of aspartate, asparagine, glutamate, and glutamine were determined by liquid-chromatography mass spectrometry (LC-MS).

E  From the same experiment as in panel (D), key metabolites involved in urea cycle, pyrimidine synthesis, TCA cycle, oxidation, glycolysis, and carnitines metabolism were determined. The NAD$^+$/NADH ratio of the indicated conditions was calculated and normalized to control (mean ± SEM). Dash line indicates indirect effect. TCA cycle, tricarboxylic acid cycle; OAA, oxaloacetic acid; UMP, uridine monophosphate; CMP, cytidine monophosphate; PEP, phosphoenolpyruvate; NADH, nicotinamide adenine dinucleotide (reduced form); NAD$^+$, nicotinamide adenine dinucleotide (oxidized form); NADPH, nicotinamide adenine dinucleotide phosphate (reduced form); NADP$^+$, nicotinamide adenine dinucleotide phosphate (oxidized form); FAD, flavin adenine dinucleotide; GSSG, glutathione disulfide; HMG-CoA, 3-hydroxy-3-methylglutaryl-CoA.

Data information: Median peak intensity was used for raw data normalization in (D) and (E). Results were calculated based on three replicates and presented as mean ± SD (unless otherwise stated). The *P*-value was calculated by two-tailed unpaired *t*-test from Prism7. \**P* < 0.05, \*\**P* < 0.01, \*\*\**P* < 0.001.

between ASNase and mock treatment (Figs 5B and EV5C). Moreover, consistent with recent reports (Garcia-Bermudez *et al*, 2018; Sullivan *et al*, 2018), implantation of SLC1A3-expressing 4T1 cells resulted in relatively faster tumor growth compared to that of parental 4T1 cells (Fig EV5D). Once tumors reached the volume of ~500 mm³, mastectomy was performed to remove the primary tumors. The amino acid analysis of the harvested tumor samples by mass spectrometry revealed almost complete depletion of asparagine by ASNase, regardless of SLC1A3 expression status, and slightly reduced aspartate levels in parental 4T1 cells derived tumors following ASNase treatment (Fig 5C). Intriguingly, we also observed depleted glutamine and glutamate levels in control tumors (Fig 5C). This might relate to the absence of SLC1A3 or other aspartate/glutamate transporters in 4T1 cells, which might decelerate glutamine replenishment in the presence of ASNase, at least in this model. Of note, the introduction of SLC1A3 into 4T1 cells (4T1-V5-SLC1A3) increased intratumor aspartate and glutamate levels, and further negated aspartate and glutamine depletions by ASNase treatment at the cost of glutamate consumption (Fig 5C).

Following mastectomy, mice survival rate was scored. In agreement with the effect of ASNase on primary tumor establishment, mice bearing tumors derived from parental 4T1 cells survived better early after ASNase treatment than mock-treated mice (Fig EV5E, left). In contrast, ASNase treatment had no effect on the survival rate of mice with SLC1A3-expressing tumors, even at early stage (Fig EV5E, right).

Recently, the bioavailability of asparagine was reported to govern breast cancer metastasis, and ASNase could reduce breast cancer metastasis (Knott *et al*, 2018). From our results above, SLC1A3-mediated aspartate/glutamate imports could affect ASNase treatment. Therefore, we next assessed whether SLC1A3 expression could negate the inhibitory effect of ASNase on cancer cell invasion in a mouse metastasis model for human breast cancer cells as described recently (Knott *et al*, 2018). For this purpose, we used MDA-MB-231 human

breast cancer cells whose metastasis burden was reduced by ASNase (Knott *et al*, 2018). Similar to 4T1 cells, MDA-MB-231 cells hardly expressed SLC1A3 (Fig 2A), did not take up aspartate, and were highly sensitive to ASNase (Fig EV5F). Consistent with above results (Fig 2G and H), SLC1A3 expression increased aspartate uptake and promoted MDA-MB-231 cell proliferation in the presence of ASNase (Fig EV5F). We therefore introduced MDA-MB-231 and MDA-MB-231-V5-SLC1A3 cells intravenously into NSG mice and assessed the invasive burdens in lung and liver. As previously reported (Knott *et al*, 2018), ASNase treatment reduced metastasis of parental MDA-MB-231 cells to the lung (Fig 5D). In contrast, the introduction of SLC1A3 increased metastatic burdens and overcame the inhibitory effect by ASNase (Fig 5D). Thus, we conclude that SLC1A3 expression induces tumor progression and ASNase resistance.

## Discussion

Although asparagine deprivation by ASNase was discovered as an effective treatment in lymphomas approximately 5 decades ago, its clinical implementation to other tumor types failed (Clarkson *et al*, 1970; Pui *et al*, 2009; Hays *et al*, 2013). The resistant mechanism to ASNase treatment in solid tumors was mainly attributed to the activation of the general amino acid sensing machinery (GCN2) and asparagine synthesis by ASNS via the GCN2-ATF4-ASNS axis (Scherf *et al*, 2000; Bunpo *et al*, 2009; Ye *et al*, 2010; Nakamura *et al*, 2018). However, the expression of ASNS in ALL did not compromise ASNase effectivity (Vander Heiden & DeBerardinis, 2017), indicating that the ubiquitous activation of the GCN2-ATF4-ASNS axis in response to nutrient deprivation might be essential, but not sufficient to induce ASNase resistance. Very recently, protein degradation was proposed to contribute to ASNase resistance in ALL (Hinze *et al*, 2019); however, its contribution in the context of solid tumors is not known yet. Here, we described the

**Figure 4. Gene expression changes pinpoint key pathways involved in SLC1A3-mediated ASNase resistance.**

A–C  PC3 (A), DU145 (B), and DU145-V5-SLC1A3 (C) cells were treated with ASNase (0.3 U/ml in A; 0.2 U/ml in B and C), TFB-TBOA (5 μM) for 3 days as indicated and subjected to transcriptome analysis. Bioinformatics pathway or gene ontology (GO) biological process analysis was performed on the sets of genes that were upregulated or downregulated when PC3 cells were treated with ASNase and TFB-TBOA compared to mock. Transcriptome analysis was based on one biological replicate for each cell line and validated by real-time PCR experiments in Fig EV4A–C. Heatmap presents row scaled normalized read counts, and the biological signaling pathway enrichment analysis was performed by ToppGene online program (Chen *et al*, 2009).

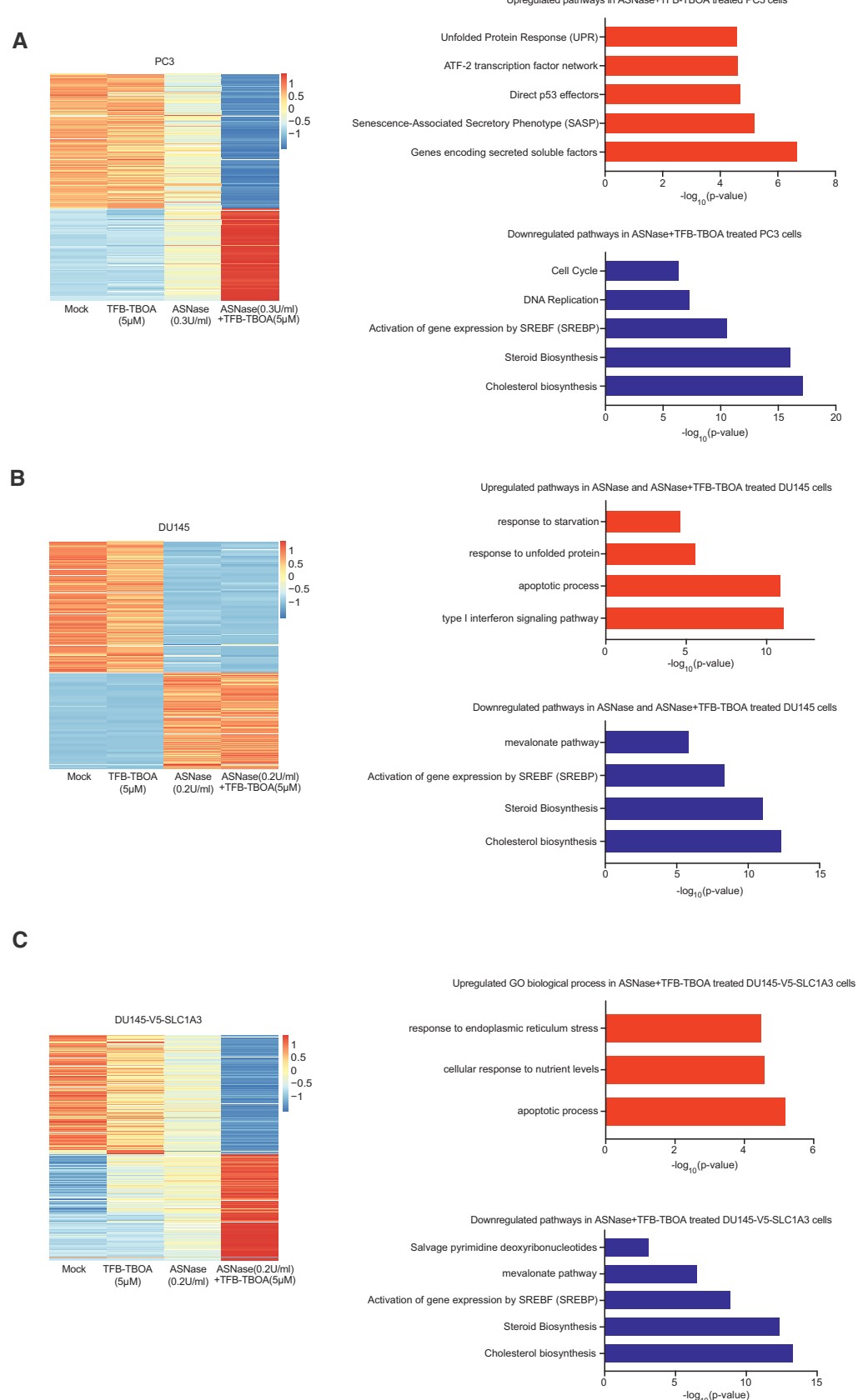

**Figure 4.**

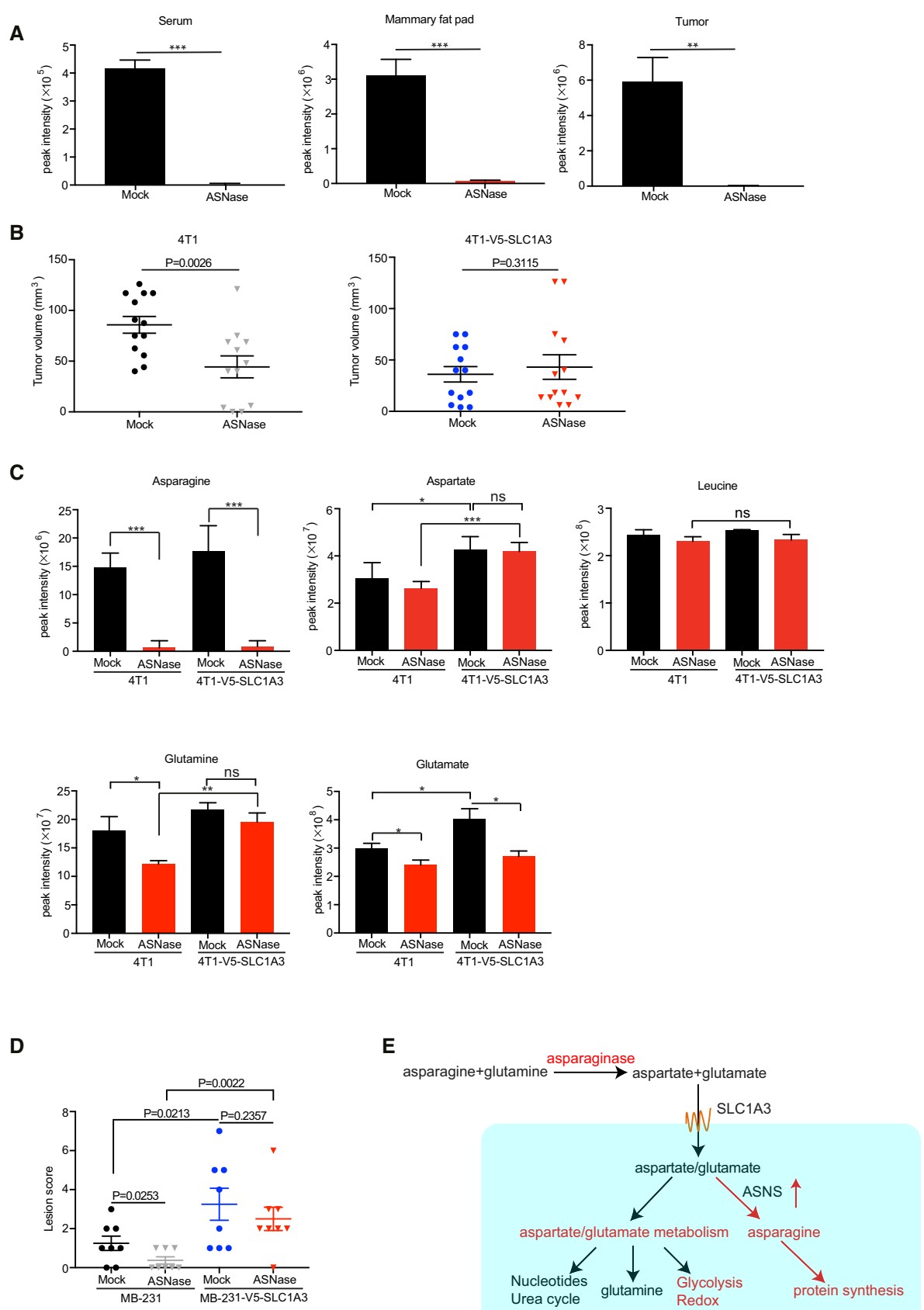

**Figure 5.**

**Figure 5. SLC1A3 expression promotes ASNase resistance and tumor progression in a mice model for breast cancer metastasis.**

A  SUM159PT human breast cancer cells were orthotopically injected into the mammary glands of NSG mice. Once SUM159PT tumors reached 250 mm³ volume, mice were treated with mock or ASNase (60 U per day) for 5 consecutive days (*n* = 3). Following treatment, mice were sacrificed, and blood, mammary glands, and tumors were collected and subjected to mass spectrometry to determine the asparagine level. Essential amino acids were used for the raw data normalization. Data are presented as mean ± SD.

B  The mouse breast cancer cell lines 4T1 and 4T1-V5-SLC1A3 were orthotopically implanted into the mammary glands of pretreated NSG mice. Presented is the volume measurements of arising tumors at day 9 (*n* = 13 mice for each group, except for 4T1 + ASNase, *n* = 12). Data are presented as mean ± SEM.

C  From the same experiment in panel (B), tumors were surgically removed once reached a volume of ~500 mm³ and collected and subjected to LC-MS to determine the levels of asparagine, aspartate, glutamine, and glutamate. Leucine level was used as a control. Results are based on five tumor samples and presented as mean ± SEM.

D  The human breast cancer cell lines MDA-MB-231 and MDA-MB-231-V5-SLC1A3 were intravenously injected into pretreated NSG mice. Once mice showed breathing problems, they were sacrificed, and lung and liver were collected and blindly scored for metastasis lesions. The *P*-value was calculated by one-tailed unpaired *t*-test in Prism7. Data are presented as mean ± SEM (*n* = 8).

E  A schematic model depicting how SLC1A3-mediated aspartate and glutamate uptake promotes ASNase resistance.

Data information: The pretreatment started 2 days before the injection of cancer cells. And mice were either injected with 60 U ASNase or saline per day. The *P*-value was calculated by two-tailed unpaired *t*-test in Prism7, unless otherwise stated. \**P* < 0.05, \*\**P* < 0.01, \*\*\**P* < 0.001.

identification of SLC1A3, an aspartate/glutamate transporter, as a novel contributor to ASNase resistance and metastasis in cancer cells. As SLC1A3 is specifically expressed in brain tissues, this expression pattern may be beneficial to guide ASNase treatment in solid tumors.

ASNase could break down both asparagine and glutamine, even though its glutaminase activity was not required for ASNS-negative cancer cells (Chan *et al*, 2014). Moreover, ASNase was found effective in treating solid tumors with intrinsic loss of ASNS (Li *et al*, 2019). We observed that in cell culture conditions, both asparagine and glutamine were robustly depleted by ASNase (Fig 1G). However, *in vivo* conditions, asparagine was far more effectively depleted than glutamine (Figs 5C and EV5A), probably due to the abundant bioavailability and timely replenishment of glutamine that reduced the effect of glutaminase activity of ASNase. The importance of asparagine to tumor cell survival was further highlighted in recent studies. Ye *et al* (2010) have demonstrated the importance of asparagine synthesis via GCN2-ATF4 axis for tumor cell survival during nutrient deprivation. And it has been demonstrated the essential role of asparagine in promoting cancer cell proliferation and breast cancer metastasis (Krall *et al*, 2016; Knott *et al*, 2018; Pavlova *et al*, 2018). Our study provides another support for the role of asparagine in cancer biology and puts forward the potential usage of ASNase in cancer therapy.

According to a previous study, aspartate metabolism was predicted to contribute to ASNase resistance in primary ALL samples (Chen *et al*, 2011). Our results that ASNase resistance could be provoked by either ectopic SLC1A3 expression or the supplementation of membrane-permeable aspartate/glutamate strongly support this hypothesis (Figs 1H, and 2H and I). It indicates that SLC1A3-mediated fueling of endogenous aspartate/glutamate levels is a novel contributor to ASNase resistance. Moreover, aspartate was the second most enriched amino acid (after asparagine) for genes related to epithelial-to-mesenchymal transition (Knott *et al*, 2018). In line with this, we observed that SLC1A3 expression could promote cancer cell metastasis, regardless of asparagine bioavailability (Figs 5D and EV5E). Even though recent studies mainly focused on the role of SLC1A3 in mediation of aspartate uptake (Alkan *et al*, 2018; Garcia-Bermudez *et al*, 2018; Sullivan *et al*, 2018; Tajan *et al*, 2018), we could not exclude the role of glutamate, which could be converted to aspartate via oxidative or reductive carboxylation. This is supported by our findings

that both aspartate and glutamate could rescue ASNase toxicity in SLC1A3 KO or negative cancer cells (Figs 1H and 2I).

Notably, we demonstrate that SLC1A3 inhibition in combination with ASNase treatment could hinder cancer cell proliferation by inducing either cell cycle arrest or apoptosis, which was observed in ALL cells following ASNase treatment (Kidd, 1953; Broome, 1961; Ueno *et al*, 1997; Pui *et al*, 2009). Metabolic and transcriptomic profiles of cancer cells treated with ASNase and SLC1A3 inhibition indicated numerous defects in many critical processes (Figs 3E and EV2F). Intriguingly, in addition to the well-known engagements in urea cycle, nucleotide synthesis, and TCA cycle replenishments (Rabinovich *et al*, 2015; Sullivan *et al*, 2015; Van Vranken & Rutter, 2015), aspartate and glutamate metabolisms might also directly or indirectly influence energy production, redox homeostasis, and lipid metabolism following ASNase treatment (Fig 5E).

Our metabolomic and "diricore" analyses indicated that ASNase-treated SLC1A3-expressing cancer cells and tumors still present asparagine shortage (Fig 5A and C; Loayza-Puch *et al*, 2016). Consistent with a previous study (Sullivan *et al*, 2018), our metabolic flux assays demonstrated that asparagine pool was not efficiently replenished by labeled aspartate (Fig EV3A). In mammalian cells, the lack of asparaginase activity prohibits asparagine utilization for the production of other amino acids or metabolic intermediates and the role of asparagine became essential when glutamine was depleted, even though it was only for protein synthesis (Pavlova *et al*, 2018).

Homologues of SLC1A3 (SLC1A1, SLC1A2, SLC1A6, and SLC1A7) can also transport aspartate/glutamate (Kanai *et al*, 2013). It remains to be further investigated whether these transporters also contribute to ASNase resistance in some cancer cells. The compound TFB-TBOA could potently inhibited SLC1A3, which leads to the negation of SLC1A3 in ASNase resistance *in vitro*. However, *in vivo* tests with TFB-TBOA showed poor pharmacokinetics activity (a sharp drop in serum levels 7 h postinjection, data not shown). Future pharmacological manipulation of TFB-TBA might be needed to improve its *in vivo* performance.

Altogether, using a genome-wide functional genetic approach, we identified SLC1A3, an aspartate/glutamate transporter, as a key determinant in the survival of cancer cells during ASNase treatment. We pinpointed the role of aspartate/glutamate in fueling metabolic pathways related to urea cycle, nucleotide, energy production, redox

homeostasis, lipid metabolism, and glutamine biosynthesis, in this process. Our results show that solid tumors are amendable to systemic administration of ASNase, opening the possibility of expanding ASNase benefit to solid tumors.

# Materials and Methods

### Cell culture

The human prostate cancer cell lines (PC3, DU145, LNCaP, and 22RV1) were bought from ATCC and cultured in RPMI (Thermo Fisher Scientific). The human breast cancer cell lines (MCF7, MDA-MB-231, and MDA-MB-468) were cultured in high glucose DMEM. SUM159PT cells (archived in the laboratory) were cultured in DMEM/F12 1:1 medium with addition of insulin (sigma, I1882-100MG, final concentration of 5 μg/ml) and hydrocortisone (sigma, final concentration of 1 μg/ml). BT549 cells (archived in the laboratory) were cultured in RPMI with insulin (final concentration of 5 μg/ml). The mouse breast cancer cell line 4T1 was a gift from O. v. Tellingen (Amsterdam, the Netherlands) and cultured in DMEM (high glucose). HEK-293T packaging cell line for lentivirus production was cultured in high glucose DMEM. All the mediums were supplemented with 10% FBS, 1% penicillin/streptomycin except for SUM159PT cells (5% FBS + 1% penicillin/streptomycin). All the cells were cultured in a humidified 37°C incubator with 5% $CO_2$ injection.

### IncuCyte cell proliferation assay

Cells were seeded in 96-well plate (Greiner, 655090), and three images per well were taken every 4 h by the IncuCyte imaging system (Essen Bioscience). Cell confluence per well was calculated by averaging the mapped areas for those three images. Experiments were performed with independent triplicates.

### Generation of SLC1A3 expression plasmid

SLC1A3 cDNA was amplified from the pLX304-SLC1A3 plasmid kindly gifted by Roderick Beijersbergen (Amsterdam, the Netherlands) using the following primer sequences: 5′-ACAGCGTCTAGA CCGGTTAGCGCTAGCTCATTAC-3′ and 5′-CGACAGTTAGCCAGAG AGCTCGCGGCCGCCGCTGT-3′. The resulting product was digested using XbaI (Roche) and NotI (Thermo Fisher Scientific) restriction enzymes and ligated into a pLenti backbone (Korkmaz *et al*, 2016) with compatible sticky ends.

### Lentivirus production and infection

A third-generation lentivirus packaging system consisting of pCMV-VSV-G (Addgene#8454), pRSV-Rev (Addgene #12253), and pMDLg/pRRE (Addgene #12252) was co-transfected with lenti-CRISPR v2 (Addgene: #52961) containing sgRNA. Transfection was performed in HEK-293T cells using PEI (polyethylenimine, Polysciences), and medium was refreshed after 18 h. Virus was harvested 48 h after transfection by snap-frozen and stored at −80°C. Target cells were incubated with virus for 24 h, and then, medium was refreshed. Thirty-six hours after virus infection,

target cells were selected with either puromycin (1 μg/ml) or blasticidine (5 μg/ml) according to the need of the experiments. The selection stopped when no surviving cells remained in the no-transduction control plate and cells were switched to normal culture medium.

### CRISPR-Cas9 genome-wide screen in PC3 cells and MAGeCK analysis

PC3 cells were transduced with lentivirus pools containing sgRNAs of a genome-wide CRISPR-Cas9 Brunello library (Doench *et al*, 2016; addgene #73179) at a multiplicity of infection (moi) of ~0.3 and ~1,000 × representations for each guide. After 2~3 days of recovery from puromycin (1 μg/ml) selection, cells were split into two different conditions: One was subjected to ASNase treatment (0.3 U/ml, ITK) for 20 days, and the other to mock treatment. Two independently replicates were included. Subsequently, genomic DNA was isolated using the phenol–chloroform extraction protocol and sgRNAs were amplified using a two-step PCR protocol for next-generation sequencing. Libraries were sequenced in an Illumina HiSeq-2500 sequencer, and raw reads were demultiplexed and analyzed using the in-house perl script XCALIBR (https://github.com/NKI-GCF/xcalibr). The individual sgRNAs abundance was further analyzed using MAGeCK (Li *et al*, 2014) pipeline to find genes statistically depleted during the screening. The MAGeCK software was ran with default options, and the 1,000 non-targeting sgRNAs included in the CRISPR-Cas9 library were used for control normalization.

Fist PCR forward primer: 5′- ACA CTC TTT CCC TAC ACG ACG CTC TTC CGA TCT NNN NNN GGC TTT ATA TAT CTT GTG GAA AGG ACG -3′ and first PCR reverse primer: 5′- GTG ACT GGA GTT CAG ACG TGT GCT CTT CCG ATC TAC TGA CGG GCA CCG GAG CCA ATT CC -3′. The forward primer contained a barcode (NNNNNN) that enabled multiplexing.

Second PCR forward primer: 5′- AAT GAT ACG GCG ACC ACC GAG ATC TAC ACT CTT TCC CTA CAC GAC GCT CTT CCG ATC T -3′ and reverse primer: 5′- CAA GCA GAA GAC GGC ATA CGA GAT CGA TGT GTG ACT GGA GTT CAG ACG TGT GCT CTT CCG ATC T -3′.

### Competitive cell proliferation assay

PC3 parental cells were stably transfected with pLKO-H$_2$B-GFP and mixed with plentiv2-sgRNA transfected PC3 cells (GFP-negative) at a ratio of ~3:7 and seeded into 12-well plates in the absence or presence of ASNase (0.3 U/ml). Cells were split every 3~5 days, and the ratio of GFP-negative cells among the mixed population was measured by flow cytometry (Calibur, BD Biosciences). GFP-negative cell counts at each timepoint were normalized to day 0 when the cells were initially mixed.

### Radioactive aspartate uptake assay

Cells were counted and seeded 1 day before the assay in 12-well plates as described (Loayza-Puch *et al*, 2017). After washed twice with PBS, cells were incubated, respectively, with [$^3$H] L-leucine (in sodium-free uptake buffer) and [$^3$H] L-aspartate (in PBS) for 5 min. Next, cells were washed twice with ice-cold PBS and collected with 0.1 M NaOH. The counts for radioactivity were measured by a

liquid scintillation analyzer on LSC2910 PerkinElmer Counter. Leucine uptake was used for normalization.

### BrdU labeling

For PC3 cells, a final concentration of 10 μM bromodeoxyuridine (BrdU, Sigma) was added to the medium and incubated for 25 min. Cells were harvested and fixed with 70% cold ethanol at 4°C for 30 min. RNase A treatment (final concentration at 0.5 mg/ml) at 37°C for 30 min was applied. Cells were resuspended in freshly prepared HCl/0.5% Triton solution (for DNA denature) at room temperature for 20 min and then neutralized by 0.1 M $Na_2B_4O_7$. After washed once with PBS/Tween, cells were incubated with 1:40 diluted anti-BrdU antibody (Dako) at RT for 30 min. Cells were incubated with FITC-conjugated anti-mouse Alexa Fluor 488 secondary antibody (1:500, Dako) at RT for 30 min in the dark. After washing another 2X, cells were then resuspended in PI (20 μg/ml) solution and ready for FACS assay (at least 10,000 cells were gated for each condition).

### Metabolite profiling and isotope tracing

$1.5 \times 10^5$ cells were seeded in 6-well plates and treated for 72 h as indicated. After washed twice with cold PBS, cells were subjected to 1 ml lysis buffer composed of methanol/acetonitrile/$H_2O$ (2:2:1) for metabolites extractions. The lysates were collected and centrifuged at 16,000 $g$ (4°C) for 15 min, and the supernatant was transferred to a new tube for further liquid-chromatography mass spectrometry (LC-MS) analysis. The LC-MS analysis procedure and parameters were used as described before (Loayza-Puch *et al*, 2017). Metabolites were identified and quantified using LCquan software (Thermo Scientific) on the basis of exact mass within 5 ppm and further validated by concordance with retention times of standards. Peak intensities were normalized based on median peak intensity of total metabolites or on essential amino acids. For Fig 5A, 10 μl of serum was diluted in 1 ml lysis buffer. For Fig 5A and C, 50~100 mg mammary fat pad tissues and tumors were ground in a mortar under liquid nitrogen, and metabolites were extracted by adding 500 μl lysis buffer and sonicated for 10 min before centrifugation.

For isotope tracing experiment, $2.5 \times 10^5$ DU145-V5-SLC1A3 cells were seeded in 6-well plates. Next day, cells were exposed to either mock or ASNase (0.2 U/ml) for 48 h and then supplemented with either 1.5 mM [$^{13}C_4$,$^{15}N$] L-aspartate (Cambridge Isotope Laboratories, CNLM-544-H) and 1.5 mM unlabeled glutamate (Sigma, G8415) or 1.5 mM [$^{13}C_5$,$^{15}N$] L-glutamate (Cambridge Isotope Laboratories, CNLM-544-H) and 1.5 mM unlabeled aspartate (Bioconnect 47203.01) for 8 h. Then, the cells and the medium were harvested for further analysis as described above.

### Total RNA isolation

Total RNA was isolated using Trisure reagent (Bioline) following the manufacturer's instructions. Briefly, cells were washed twice with PBS and 1 ml Trisure was added for homogenization. After centrifuge, the aqueous phase was transferred to a new tube and mixed with cold isopropyl alcohol for RNA precipitation by centrifuging at 4°C for 1 h. RNA pellet was washed twice with 75% ethanol and finally dissolved in RNase-free water.

### Reverse transcription and quantitative real-time PCR (RT–qPCR)

Reverse transcription was performed with Tetro Reverse Transcriptase kit (Bioline) according to the manufacturer's instructions. Briefly, 2 μg of total RNA was used as templates for each reaction. qPCR products were prepared using a SensiFAST SYBR No-ROX kit (Bioline) according to the instructions and performed in the Light Cycler 480 (Roche). Primers are listed in Table EV1.

### Western blot analysis

Cells were washed twice with PBS and lysed with 2 × SDS buffer (4% SDS, 20% glycerol and 125 mM Tris PH 6.8). Next, protein levels were quantified by Pierce BCA protein assay kit (Thermo Scientific). Lysates were loaded into a separating 7.5% SDS–PAGE gel, and protein was transferred to nitrocellulose membranes. After blocking with 5% milk/PBS–Tween-20 (0.2%) solution, the membrane was incubated with mouse-anti-V5 (Thermo Fisher Scientific). Proteins were visualized using the secondary fluorescently-labeled antibodies goat anti-mouse IRDye 680 RD (LI-COR Biosciences) and scanned on the Odyssey infrared imaging system (LI-COR Biosciences).

### Immunofluorescence assay

Cells were grown on glass coverslips, washed twice with PBS, and fixed with 2% PFA for 10 min at room temperature. Next, cells were permeabilized with 0.5% Triton/PBS solution, blocked with 5% FBS for 1 h, and incubated with mouse-anti-V5 (Thermo Fisher Scientific) and Alexa-488-conjugated rabbit anti-mouse secondary antibodies. Coverslips were mounted on glass slides using Vectashield containing DAPI. Images were taken with Leica confocal microscope SP5.

### TruSeq standard mRNA sample preparation

Stranded-specific libraries were generated using the TruSeq Stranded mRNA sample preparation kit (Illumina) following the manufacturer's instructions. Briefly, 2 μg of total RNA was polyA-selected using oligo-dT beads and the RNA was fragmented, random primed, and reverse transcribed using SuperScript II Reverse Transcriptase kit (Invitrogen). Second-strand complementary DNA was then synthesized, 3′-adenylated and ligated to Illumina sequencing adapters, and subsequently amplified by 12 cycles of PCR. The sequencing libraries were analyzed on a 2100 Bioanalyzer using a 7500 chip (Agilent) and pooled equimolarly into a 30 nM multiplex sequencing pool.

### Deep sequencing

Samples were sequenced on the Illumina HiSeq2500 sequencer generating 65-nucleotide single-end reads.

### RNA-seq analysis

Sequenced reads were aligned to the human genome (hg19) using TopHat v2.0.8 (Trapnell *et al*, 2009). Only uniquely mapped reads were retained for further analysis. SAMTOOLS v0.1.19 (Li *et al*,

2009) was used to convert the BAM output to SAM format and to sort the BAM file. The read counts per gene were calculated using the HTSeq program, v0.5.4p1 (Anders *et al*, 2015). The DESeq package (Oshlack *et al*, 2010) was used to generate normalized read counts and for differential gene expression analysis. DESeq called differentially expressed genes with FDR cutoff of 0.05 and abs (FC) > 1.5 were considered as significantly differentially expressed genes.

### IncuCyte® Caspase-3/7 Green apoptosis assay

Cells were pre-seeded in 96-well plate (Greiner, 655090) 48 h before the addition of Caspase-3/7 Green Apoptosis reagent (Essen Bioscience, 4440). The green signals were captured every 4 h, and apoptotic cells were counted.

### Animal studies

All mice experiments were approved by the Netherlands Cancer Institute Animal Experimental Committee. For Fig EV1E, xenografts were induced by subcutaneous injection of $4 \times 10^6$ PC3 control (sgNon-targeting) and sgSLC1A3 cells (monoclonal #4-1) in one flank of Balb/c nude mice ($n = 8$) and treatment started when tumors reached 50 mm³. For Fig 5A, $4 \times 10^6$ SUM159PT cells were resuspended in 50 µl PBS and injected into the mammary gland #4 of NOD-SCID IL2R-null (jax) (NSG) mice. After tumor volumes reached ~250 mm³, mice ($n = 3$ per group) were administrated either with mock (saline) or ASNase (60 U per day) for 5 consecutive days by intraperitoneal injections. Serum, mammary fat pad tissues, and tumors were collected and snap-frozen for LC-MS analysis when the mice were sacrificed. For Fig 5B, $1 \times 10^5$ 4T1 cells and 4T1-V5-SLC1A3 cells were respectively resuspended in 20 µl 1:1 mix of PBS and growth factor reduced Matrigel (Geltrex™, Gibco) and injected into 1 mammary fat pad per mouse. Mice were pretreated for 2 days either with saline or ASNase (60 U per day) before tumor cells were introduced, and the treatment was performed every day until the mice were sacrificed ($n = 13$ per group except for the group of 4T1 treated with ASNase, $n = 12$). Primary tumors were surgically removed once the tumor volumes reached 450–550 mm³, and mice underwent breathing challenges every day. For Fig 5D, $5 \times 10^4$ MDA-MB-231 and MDA-MB-231-V5-SLC1A3 cells (in 50 µl PBS) were, respectively, injected into the tail veins of 2 days of pretreated NSG mice ($n = 8$ per group) and mice were sacrificed 22 days post-tumor cells introduction.

All the experiments were using NSG mice 6~8 weeks old (except for Fig EV1E), and mice were weighed every 2 or 3 days to monitor weight loss. For ASNase treatment, mice were intraperitoneally administrated 60 U ASNase every day till the end of the experiments. Tumor volumes were calculated by the formula $V = 1/2(LW^2)$, where $L$ is length and $W$ is width of the primary tumor.

### Histopathology analysis of lung and liver invasion

Lung and liver tissues were collected and fixed in formalin fixative and embedded in paraffin. The immunohistochemistry (IHC) of vimentin (DAKO, M0725, dilution: 1:4,000) was conducted on 4-µm-thick sections according to standard procedures. The stained slides were examined blindly by a pathologist, and the number of tumorous lesions (more than 10 cancer cells) was scored in each of the sections. The sections were reviewed with a Zeiss Axioskop2 Plus microscope (Carl Zeiss Microscopy, Jena, Germany), and images were captured with a Zeiss AxioCam HRc digital camera and processed with AxioVision 4 software (both from Carl Zeiss Vision, München, Germany).

### Statistics

Data analyses were performed using GraphPad Prism (version 7). The statistical tests used are described in figure legends. $*P < 0.05$, $**P < 0.01$, $***P < 0.001$. For the mass spectrometry analysis of amino acids in tumor samples, no statistics methods were used to predetermine sample size. For animal experiments, an estimate was made for the number of mice needed, without power calculation.

### TCGA datasets analysis

Expression data from tumor and normal tissue samples were downloaded for every project available at ICGC data portal (http://dcc.icgc.org; release 27). For consistency, only expression data from pipeline "RNASeqV2_RSEM_genes" were considered. The downloaded normalized expression data were scaled to TPM (transcripts per million reads) and log2 transformed. Only projects with more than 10 normal samples were considered. All analyses were done using R-language. The statistical comparison between normal and tumor samples was done using a Wilcoxon sum rank test with Bonferroni correction for multiple comparisons.

## Data availability

The deep sequencing datasets generated in this study have been deposited in GEO database under accession number: GSE134074 (https://www.ncbi.nlm.nih.gov/geo/query/acc.cgi?acc=GSE134074). All other data generated that support the findings of this study are available from the corresponding author upon reasonable request.

Expanded View for this article is available online.

### Acknowledgements

We acknowledge all the members in Agami laboratory for valuable discussions. We thank Roderick Beijersbergen for providing the CRISPR-Cas9 library plasmid; Ron Kerkhoven, Roel Kluin, and Iris de Rink from the NKI Genomics Core Facility for assisting with deep sequencing experiments and analysis. We appreciated the assistance from NKI Experimental Animal Pathology, Radionuclides Center, and Flow cytometry facilities. We also thank the people from the Preclinical Intervention Unit and Pharmacology Unit of the Mouse Clinic for Cancer and Ageing (MCCA) at the NKI for performing the intervention studies. This work was supported by funds of the China Scholarship Council (201503250056) to J.S., the Dutch cancer society (KWF 0315/2016), and the European research council (ERC-PoC 665317) to R.A.

### Author contributions

JS and RA conceived the project, designed the experiments, and wrote the manuscript. JS performed most of the *in vitro* experiments for this project. RN

provided technical support and established the protocols for the *in vivo* experiments. APU performed the bioinformatics analysis. EAZ and CRB performed metabolic mass spectrometry experiments and analyzed the data. RH helped with the RNA-seq library preparation. AP analyzed RNA-seq data. AB and OT prepared materials for mice experiments. NP and MV performed the *in vivo* experiments. HF and GJP helped with the synthesis of TFB-TBOA. J-YS performed histopathologic analysis. The project was supervised by RA.

## Conflict of interest

The authors declare that they have no conflict of interest.

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
