## [Review Process File · The EMBO Journal]

SLC1A3 contributes to L-asparaginase resistance in solid tumors

Jianhui Sun, Remco Nagel, Esther A. Zaal, Alejandro Piñeiro Ugalde, Ruiqi Han, Natalie Proost, Ji-Ying Song, Abhijeet Pataskar, Artur Burylo, Haigen Fu, Gerrit J Poelarends, Marieke van de Ven, Olaf van Tellingen, Celia R. Berkers, Reuven Agami

Review timeline:

Submission date:	29th Mar 2019
Editorial Decision:	7th May 2019
Revision received:	14th May 2019
Editorial Decision:	24th May 2019
Revision received:	1st Jul 2019
Editorial Decision:	9th Aug 2019
Revision received:	13th Aug 2019
Accepted:	14th Aug 2019

Editor: Daniel Klimmeck

Transaction Report:

1st Editorial Decision

7th May 2019

Thank you again for submitting your manuscript for consideration by The EMBO Journal and your interest. Your manuscript has now been seen by three referees whose comments are shown below. In light of these comments, I am afraid we decided that we cannot offer publication in The EMBO Journal. I do however encourage you to consider transfer of this work to our sister journal EMBO Reports.

As you can see, the referees appreciate that the analysis extends previous work and state overall interest in the results. However, they also raise major concerns with the analysis that I am afraid preclude publication here. Referee #1 states that your claims on a broader pathophysiological relevance of SLC1A3 in solid tumors are not fully convincing in light of negative cancer entities. In addition this referee expresses major concerns, that glutamate dependency of the observed phenotypes is not addressed but could be an important confounding factor (ref#1, pt.4). Both referees #2 and #3 agree in that roles of glutamate transport by SLC1A3 in sensitivity are not considered, which in their view undermines the impact of the findings (ref#2, pt.2; ref#3, pt. 1). Referee #2 finds that aspartate availability has been linked to metastasis before, and referee#3 states that the functional implication SLC1A3 is rather unclear in his/her view. In addition, the referees state issues related to data display and experimental setup, consistency between cell lines as well as controls missing and they request essential experiments to support and expand the concept made.

Given these critical opinions from good experts on the field, and considering the huge amount of revisional work requested in light of the single major round of revisions we usually offer as to our journal policy, I am afraid we cannot offer to publish your work here.

REFeree REPORTS:

Referee #1:

The authors' goal was to identify the mechanism for solid tumours' resistance to L- asparaginase. They performed a crisper screen on prostate cancer cells and identified 3 confirmed genes which gave synthetic lethality with L-Asparaginase treatment. KO of these genes by itself did not affect proliferation. Out of the 3, SLC1A3 was the most significant and yet unexplored in connection to L- asparaginase. SLC1A3 is an aspartate and glutamate transporter.

Following asparaginase treatment, aspartate levels depleted further in the SLC KO cells suggesting higher consumption of intracellular aspartate.

Esterified aspartate rescued proliferation in SLC KO treated with asparaginase. The researchers tested several breast and prostate cancer cells and found correlation between SLC1A3 expression and aspartate uptake, when there was no expression of other aspartate transporters. OE of SLC1A3 rescued prostate and breast cancer cells with SLC1A3 deficiency and conferred resistance to asparaginase. They further prove the dependence on SLC1A3 using a chemical inhibitor. With isotope tracing the authors show that following asparaginase treatment, the metabolic fates of cellular aspartate remains the same. Transcriptomic analysis supports the metabolic data for cancer cells treated with asparaginase with and without a functional SLC1A3. The main conclusion of the authors is that SLC1A3 provides resistance to aspartate in solid tumors and hence limiting aspartate intake might improve aspartate treatment efficacy.

General comments- the authors did a lot of work aiming to show the importance of SLC1A3 in providing resistance to asparaginase treatment in vitro and in vivo. Overall, I believe their conclusion is likely to be correct. However, there are major issues that need to be addressed to make it factual. First, structurally, the paper has a problem in referring to the correct figures, which make it very hard to follow. Some figures are not even referred to and others are mis- referred. The figures themselves are hard to understand from the legends, which are very minimally detailed. There also seems to be pick and choose of what to show in these figures. Together, it gives an impression of a sloppy approach that weakens the data.

Examples- EV2 is not referred, should it be in line 106 replacing the reference to Fig 2A?

Figure 4 has A-D and not A-G (line 209) and there is no TFB-TBOA data in figure 4.

Figure legend 1 goes from C to D to C again.

2. Currently ALL patients respond to asparaginase robustly. What is SLC1A3 status in ALL? This is very basic and required to differentiate ALL from asparaginase resistant solid tumors.

3. What is SLC1A3 expression in solid tumors- most solid tumors do not express it and so why are they resistant to asparaginase? Is it upregulated following treatment? This is an important point as the authors might have identified a potential resistant mechanism that doesn't occur spontaneously in reality. The needs to be checked in the TCGA dataset.

4. The authors need to test the cancer cells for glutamate dependency following perturbations in SLC1A3 as it is also the transporter for glutamate. Glutamate can enter the cells via other transporters and can be converted to aspartate and rescue the cells.

Figure 1E as referred in the text (line 134) does not show how glutamate depletion affects proliferation in SLC KO treated with asparaginase. There is also no measurement of glutamate uptake which is critical to evaluate in the SLC KO with and without asparaginase treatment. Also, did they try to do a rescue with esterified glutamate? They discuss it in the discussion (338) but do not show this result.

Specific comments:

Figure 1: what is sgNT? I understand it is a control but there is no de-abbreviation for it anywhere in the text. The legend of figure 1 does not match the panels.

2D - no consistency in the correlation with the expression of SLC1A3 and sensitivity to asparaginase. In general, it is hard to interpret the figure because for each panel a different concentration of asparaginase was used.

Figure 3- The authors need to show specificity of TFB-TBOA for SLC1A3. Broadly, it could have multiple targets. Specifically, it could be inhibiting aspartate SLC25A13 causing depletion in

mitochondrial aspartate transport to the cytosol. This could also explain the depletion in NADH since this transporter participates in the malate aspartate shuttle.

B-C- It is hard to evaluate the comparison between the cells since the authors show cell cycle analysis for one cell line - PC3 and apoptosis analysis for the other two. Both assays should be shown for all 3 cell lines, preferably using the same method- FACS analysis that includes an antibody for apoptosis as PARP. Also, I would expect nucleotide shortage to induce cell cycle arrest in G1 and not S. Please explain the result.

3E- In addition, the authors need to show that the urea cycle enzymes are expressed in the cancer cells they are using as these are often not expressed in tumours.

EV3- All staining needs to be quantified.

Figure 4: The authors need to perform tracing with glutamine because it is the major TCA replenishing metabolite that also contributes to urea cycle intermediates and pyrimidine synthesis.

Figure 5- It is unclear which perturbation is compared to which in terms of pathway analysis. Is there an explanation the authors have as to why A and C with double treatment look opposite? I would expect them to share mechanisms since they both express slc1a3. Also, upregulation of p53 as is seen in A should support a G1 or G0 arrest and not a prolonged S phase as is shown in figure 3.

Figure 6- 4T1 can be injected to immune competent mice (BALB/c). It would make the data more relevant since the results will factor in the role of the immune system in asparaginase efficiency. For consistency, they can use 4T1 with luciferase to evaluate metastasis formation in a mouse with functional immune system.

Referee #2:

Summary: Sun et al. characterize potential resistance mechanisms to L-asparaginase treatment in solid tumors utilizing a Crispr/Cas9 based approach in the PC3 prostate cancer cell line. Their genetic screen highlights the expression the glutamate/aspartate transporter SLC1a3 as a possible resistance mechanism. The authors show that genetic or pharmacologic inhibition of SLC1a3 increases sensitivity to L-asparaginase treatment in several cell lines and over-expression can confer resistance. Additionally, the authors show that SLC1a3 expression in breast cancer cells promotes tumor growth and progression alone and when treated with L-asparaginase.

The basic findings of this study are compelling and important to the field of cancer biology and cancer metabolism, however this work needs further development and requires significant revisions.

Revisions:

1. In Figure 1E it appears that all of the cell line variants (control, sgSLC1a3, ASNase treated, sgSCL1a3/ASNase treated) proliferate at roughly equivalent rates up until day 3 and then diverge, with ASNase treated and sgSLC1a3/ASNase treated cells proliferating more slowly. Asparagine and aspartate levels in these conditions are shown after 3 days of treatment with sgSLC1a3 and ASNase treatment decreasing both as expected (Figure 1G).

a. Does it take 3 days for cells treated with ASNase and sgSLC1a3/ASNase 3 days to deplete intracellular asparagine/aspartate stores, which slows proliferation? Or do control and sgSLC1a3 cells upregulate ASNS expression after 3 days and increase biosynthesis of asparagine to boost proliferation rates in comparison?

b. What do intracellular levels of asparagine and aspartate look like before and after 72hrs?

2. Over all the authors only focus on SLC1a3's ability to transport aspartate as the primary mechanism by which SLC1a3 expression confers resistance to ASNase treatment. However, SLC1a3 also transports glutamate, which is a necessary nitrogen donor to synthesize asparagine from aspartate. It is likely that intracellular glutamate levels also play a large role in dictating resistance or sensitivity to ASNase treatment.

a. Figure 1F - How is glutamate uptake effected?

- b. Figure 1G - what do glutamate levels look like in these cell lines and after treatment with ASNs?
 - c. Figure 1H - Does glutamate supplementation also rescue sensitivity to ASNase treatment? As aspartate supplementation doesn't fully rescue proliferation defects after ASNase treatment it is likely that increasing glutamate levels would provide an additional growth benefit. Does glutamate + aspartate supplementation fully rescue ASNase treatment?
 - d. Figure 2J - Does glutamate alone or in combination with aspartate rescue growth after ASNase treatment in DU145 cell line?
 - e. Figure 3D - how does the SLC1a3 inhibitor TFB-TBOA effect glutamate levels?
3. Figure 1F - the western blot should be improved or omitted from the figures as it is not does not show convincing evidence of strong over expression of SLC1a3 in these cells.
4. The manner in which the data is represented in Figure 4 is overly complicated for a main figure and difficult to interpret even if the reader has knowledge/experience with this type of experiment and analysis.
- a. Stacked graphs should be moved to supplemental figures and percent enrichment for the most relevant isotopomer for each metabolite should be shown for carbon and nitrogen individually.
 - b. The DU145 cell line used in this analysis is cultured in RPMI, however for the labeling experiment the authors use 1.5mM C13/N15 glutamine and 1.5mM C13/N15 aspartate which are not consistent with the concentrations of these amino acids found in RPMI. Why are different concentrations chosen for this experiment? Using different concentrations of these two metabolites for this experiment makes this data not directly relatable to the other experiments with this cell line throughout the paper.
 - c. Figure 4a shows that cells secrete a significant amount of aspartate which increased by ASNase treatment. Can the authors comment on why this is occurring?
 - d. This figure shows that extracellular aspartate does not contribute to the asparagine pool. If cells are exclusively relying on endogenously produced sources of asparagine why does ASNase treatment have an effect? And why does SLC1a3 inhibition increase this effect if its primary function is to import aspartate? There is no difference in labeling between control and ASNase treated cells in any of the other detected metabolites again begging the question of how aspartate supplement is boosting proliferation of cells treated with ASNase. This further points to the fact that glutamate uptake mediated by SLC1a3 may play an important role in this system.
5. There is an additional supplemental figure, EV6 that is not referenced in the text. All references to Figure EV7 are incorrect in the text and mistakenly refer to EV6.
6. Figure 6a - The Sum159PT cell line is only used to show asparagine levels in the serum, fat pad and tumor. Does this cell line respond to ASNase treatment in vivo? This data would be better if produced from animals with 4T1 tumors as this cell line is used to generate the remainder of the in vivo functional data.
7. Figure 6b - only a single time point is shown for response to ASNase treatment. How do these tumors respond over time and what do the growth kinetics look like with ASNase treatment.
- a. The authors performed primary tumor resection and survival studies using the 4T1 model which readily metastasizes. Did the authors collect lungs from these animals to perform histological analysis of metastatic lesions? This would be excellent complimentary data to the experimental metastasis model with MDA-MB231 and would show ASNase effect on spontaneous metastasis.
8. Overall the figures need more descriptive labels so that it is easy for the reader to understand what cell lines, treatments, or conditions are being represented without having to refer the text or the figure legends

Referee #3:

In this manuscript by Sun et al., the authors describe a novel role for SLC1A3-mediated aspartate uptake in resistance of solid tumors to L-asparaginase treatment. By performing a whole genome CRISPR-Cas9 screen in a prostate cancer cell line, the authors find that one of the genes whose knockout sensitizes cells to ASNase treatment is plasma membrane glutamate/aspartate transporter SLC1A3. After validating that the increase in intracellular aspartate levels mediated by expression of this transporter is responsible of ASNase resistance in many different cells by genetics and pharmacologic means, the authors engage in showing the relevance of this resistance in vivo. Knockout of SLC1A3 in prostate cancer cells impairs tumor growth upon ASNase treatment, and concomitant overexpression of the transporter in a breast cancer cell line that do not express basal SLC1A3 enables these tumors resistant to the treatment both in xenograft and orthotopic models.

Intriguingly, the authors find that overexpression of SLC1A3 strongly increases metastatic burden in the same breast cancer model, in agreement with recent studies showing that ASNase treatment and asparagine availability have an impact in metastasis. Of note, the authors provide a lot of useful data, such as metabolic changes or transcriptional responses upon ASNase treatment; and validate their results in different cancer types.

Major points:

1. ASNase is known to break down not only asparagine but also glutamine at certain concentrations. This is a critical point not addressed throughout the manuscript, since SLC1A3-mediated uptake of glutamate would also protect against glutamine depletion. Because of this, repeating some experiments and measuring some extra metabolites is recommended. A) Treat PC3 cells in vitro with the L-ASNase concentration used in your first assays (0.3-0.5 U/mL) and measure not only depletion of asparagine but also glutamine in the media. B) Validate that SLC1A3 knockouts sensitize cells to asparagine depletion by using an ASN-depleted media instead of ASNase treatment. ASN depletion by this means should phenocopy ASNase treatment effect bypassing any potential depletion of Glutamine. C) It is striking that SLC1A3 KOs show such a decrease in intracellular glutamine levels (Fig. EV1C), raising the possibility that SLC1A3 rescue of ASNase treatment could be more related to glutamine depletion than to asparagine, at least in the model. Could the authors repeat these experiments of proliferation in a media with asparagine and aspartate, but not with glutamate in it? Given the almost identical structure of glutamate and aspartate, glutamate (first product after glutaminase reaction) can be taken up by SLC1A3 and would definitely rescue GLN-deprivation phenotype. Conversely, do the same experiments in the presence of asparagine and glutamate, but not aspartate, in the media. D) In the in vivo experiments shown in Fig. 6 (Fig. 6A and 6C) with analysis of orthotopic tumors metabolite levels, could the author provide the levels of Glutamine and Glutamate?. This would help to see if Gln is depleted in the tumor and if SLC1A3 may rescue by taking up glutamate. E) The authors mention in the discussion that addition of esterified Glutamate also rescues ASNase treatment. Could the authors show these results adding it to Fig. 1H and discuss the possibilities further?
2. Could the authors comment why they think SLC25A1 scores? This is a mitochondrial citrate carrier that due to its function is required for reductive carboxylation (Jiang et al., 2016, Nature; PMID: 27049945). Is glutamine metabolism rewired from oxidative to reductive carboxylation during ASNase treatment?
3. In Figure 3E, the authors stress that NADH levels decrease after ASNase treatment, and that this is indicative of a defective aspartate synthesis/metabolism. Even though this result would make sense, it is hard to believe that NADH levels decrease that much in the absence of appreciable changes in NAD⁺ levels after such a short treatment. NADH should be converted into NAD⁺ and the results shown in the figure suggest that the total pools change. The authors should show the results as NAD⁺/NADH ratio as previously done in similar studies, and probably repeat it in the presence of ASNase or the double treatment but using a fluorescent detection kit instead of mass spectrometry.
4. The results shown in Figure 4 are intriguing. Imported isotope-labeled aspartate barely labels any metabolite at appreciable levels (including pyrimidines), but still rescues ASNase sensitivity without replenishing the asparagine pool. This result suggests that, at least in the solid tumors studied here, ASNase treatment does not affect tumor growth by depleting asparagine, but rather by depleting aspartate as a result of an increased demand of asparagine synthesis from aspartate. If that's true, supplying downstream metabolic outputs of aspartate, and not asparagine, should rescue sensitivity to ASNase. Could the authors for example supplement high levels of nucleosides (both purine and pyrimidine precursors, for which aspartate is required) and rescue ASNase treatment or ASN-depletion from the media? This should work at least in PC3 cells where an impaired cell cycle is observed, which could be due to a shortage in nucleotide precursors.

Minor points:

1. In line 208-209, the authors refer the text to Fig. 4A-G but it should be Fig. EV4A-D.
2. EV6 is not referenced. EV7 is referenced as EV6 in line 260.

Thanks for your care and for considering our manuscript for publication in EMBO. I was looking carefully at the reports of the referees and I am certain that we can reply to all raised issues and

convince them about the solidity of our work. If you allow us, we will correct all text issues and provide the necessary experimental evidence that will address all referee's concerns. Please consider that two of the referees (#2 and #3 are acknowledge the importance of our work and its novelty, and are thus positive). Referee #1 was (rightly) more annoyed by the mistakes we had in the text (of which I apologize for), but also had very good points that we can fully address.

Please find below my reply to your concerns, and in the attached file point-to-point reply to the comments of the referees.

As you will see, we can (rather easily) address all the comments of the reviewers (and even have almost all the experiments in hand).

In light of this, I would like to request you to reconsider your negative decision and allow us to come back with a revised manuscript.

--

As you can see, the referees appreciate that the analysis extends previous work and state overall interest in the results. However, they also raise major concerns with the analysis that I am afraid preclude publication here. Referee #1 states that your claims on a broader pathophysiological relevance of SLC1A3 in solid tumors are not fully convincing in light of negative cancer entities.

Reply: Due to a mistake in labeling from our side, the reviewer had regrettably overlooked our results presented in EV2 (which shows the over expression of SLC1A3 in solid tumors). Together with the animal experiments that we present, I have no doubt that a revised manuscript will convince the reviewer of the broader pathophysiological relevance of SLC1A3 in solid tumors.

In addition this referee expresses major concerns, that glutamate dependency of the observed phenotypes is not addressed but could be an important confounding factor (ref#1, pt.4).

Reply: We agree with the reviewer that glutamate might be important and indicated it in the discussion of our manuscript (line 332). Due to the direct correlation of aspartate and asparagine and the increasing interest in SLC1A3's role in aspartate transportation (Sullivan et al, 2015; Garcia-Bermudez et al, 2018; Alkan et al, 2018; Sullivan et al, 2018; Tajan et al, 2018), and because aspartate and glutamate could be converted to each other, we did not include these data. In the revised manuscript we will include the results we collected about glutamine and glutamate, as requested. We will also add a discussion on this important issue.

Both referees #2 and #3 agree in that roles of glutamate transport by SLC1A3 in sensitivity are not considered, which in their view undermines the impact of the findings (ref#2, pt.2; ref#3, pt. 1).

Reply: Indeed. In the revised manuscript we will include the results we collected about glutamine and glutamate, as requested.

Referee #2 finds that aspartate availability has been linked to metastasis before,

Reply: Referee #3 mentioned the correlation of asparagine and metastasis as described by (Knott et al, 2018). In addition to confirming this finding, the results we present here show that SLC1A3 expression is also causally linked to metastasis.

and referee#3 states that the functional implication SLC1A3 is rather unclear in his/her view.

Reply: Referee#3 proposed constructive experiments that we can defiantly address within the restricted time of a revision.

In addition, the referees state issues related to data display and experimental setup, consistency between cell lines as well as controls missing and they request essential experiments to support and expand the concept made.

Reply: I deeply apologize for the mistakes in pinpointing figures. This will be fully corrected in a revised version. However, clearly the reviewers were very interested in the concept we propose, and indeed raised suggestions, that we accept and experimentally address in full.

In light of these points, I would like you to consider your decision and allow us to rebuttal. If so, I foresee that we will be able to comeback with an appropriate response within few weeks. I do hope you would agree.

I enclose a point-to-point draft of our response to the comments and suggestions of the referees.

2nd Editorial Decision

24th May 2019

Thank you for contacting me regarding our decision and for your patience with my response, which got delayed due to detailed internal discussions in the team and getting back to the referees regarding your point-by-point response.

We appreciate your outline for a substantive experimental revision, and realise that you would - judging from the information provided in the point-by-point letter - be potentially able to address the issues raised by the referees.

We thus invite you to work towards a re-review and will be able to return a revised version to the referees for evaluation. Please note however that it would be essential to address all experimental and presentation shortfalls in compelling manner and to the satisfaction of the referees. We reiterate that our sister venue EMBO Reports is also interested in the work in principle, yet, clearly a more definitive dataset will be required. Given the referees' somewhat ambivalent response, we cannot at this stage guarantee that enthusiasm for EMBO Journal will be sufficiently high based on the continuing questions around clarification of the distinct roles and importance of aspartate versus glutamate, and the specific details of SLC1A3's function in resistance. We may therefore recommend EMBO Reports in light of the referee feedback, albeit in that case without additional peer review.

2nd Revision - authors' response

1st Jul 2019

We would like first to thank the reviewers for their very constructive and useful comments and suggestions, which significantly improved our manuscript. In the revised manuscript, we experimentally addressed the comments, and implemented the required changes to the text. Below is a detailed response to the comments of all reviewers.

Referee #1:

The authors' goal was to identify the mechanism for solid tumours' resistance to L- asparaginase. They performed a crisper screen on prostate cancer cells and identified 3 confirmed genes which gave synthetic lethality with L-Asparaginase treatment. KO of these genes by itself did not affect proliferation. Out of the 3, SLC1A3 was the most significant and yet unexplored in connection to L-asparaginase. SLC1A3 is an aspartate and glutamate transporter.

Following asparaginase treatment, aspartate levels depleted further in the SLC KO cells suggesting higher consumption of intracellular aspartate.

Esterified aspartate rescued proliferation in SLC KO treated with asparaginase. The researchers tested several breast and prostate cancer cells and found correlation between SLC1A3 expression and aspartate uptake, when there was no expression of other aspartate transporters. OE of SLC1A3 rescued prostate and breast cancer cells with SLC1A3 deficiency and conferred resistance to asparaginase. They further prove the dependence on SLC1A3 using a chemical inhibitor. With isotope tracing the authors show that following asparaginase treatment, the metabolic fates of cellular aspartate remains the same. Transcriptomic analysis supports the metabolic data for cancer cells treated with asparaginase with and without a functional SLC1A3. The main conclusion of the authors is that SLC1A3 provides resistance to aspartate in solid tumors and hence limiting aspartate intake might improve aspartate treatment efficacy.

General comments- the authors did a lot of work aiming to show the importance of SLC1A3 in providing resistance to asparaginase treatment in vitro and in vivo. Overall, I believe their conclusion is likely to be correct. However, there are major issues that need to be addressed to make it factual. First, structurally, the paper has a problem in referring to the correct figures, which make it very hard to follow. Some figures are not even referred to and others are mis- referred. The figures

themselves are hard to understand from the legends, which are very minimally detailed. There also seems to be pick and choose of what to show in these figures. Together, it gives an impression of a sloppy approach that weakens the data.

Reply: We appreciate the comments the referee has made to help us refine our manuscript.

- (1) Indeed, there were some editing mistakes in referring to the corresponding figures: Fig EV4 (but not Fig 4) and Fig EV2A (but not Fig 2A). In the current manuscript we corrected all textual flaws.
- (2) We apologize for the shortcoming in the description of the figures in the legends. In the revised manuscript we have done our best to make the legends as clear as possible.

Examples- EV2 is not referred, should it be in line 106 replacing the reference to Fig 2A?

Reply: Indeed, there was EV missing. We replaced Fig 2A with Fig EV2A.

Figure 4 has A-D and not A-G (line 209) and there is no TFB-TBOA data in figure 4.

Reply: Indeed, there was EV missing. We replaced Figs 4A-G with Figs EV4A-G.

Figure legend 1 goes from C to D to C again.

Reply: The second (C) was mentioned for the explanation of (D). Each legend was paragraphed to be easily read.

2. Currently ALL patients respond to asparaginase robustly. What is SLC1A3 status in ALL? This is very basic and required to differentiate ALL from asparaginase resistant solid tumors.

Reply: We thank the reviewer for this important question. To examine this issue, we analyzed datasets from Oncomine database. Indeed, SLC1A3 expression is extremely low in ALL compared with other tumors and tissues. In addition, the same conclusion is drawn when we analyzed the database from the Broad Institute Cancer Cell Line Encyclopedia (CCLE). Thus, the expression levels of SLC1A3 in ALL is extremely low and this feature can separate sensitive ALL from resistant solid tumors.

3. What is SLC1A3 expression in solid tumors- most solid tumors do not express it and so why are they resistant to asparaginase? Is it upregulated following treatment? This is an important point as the authors might have identified a potential resistant mechanism that doesn't occur spontaneously in reality. The needs to be checked in the TCGA dataset.

Reply: We thank the referee for the comment. We present in the revised manuscript Fig EV2B, an analysis of TCGA database which indicates that high SLC1A3 expressions appears in some tumor types, especially kidney renal clear cell carcinoma (KIRC, $p = 5.5 \times 10^{-30}$), kidney renal papillary cell carcinoma (KIRP, $p = 2.1 \times 10^{-10}$), liver hepatocellular carcinoma (LIHC, $p = 3.2 \times 10^{-10}$) and stomach adenocarcinoma (STAD, $p = 6.1 \times 10^{-5}$). This high expression might indicate resistance if ASNase was applied to those tumor types.

In addition, very recent publications (Garcia-Bermudez et al, 2018; Alkan et al, 2018; Sullivan et al, 2018; Tajan et al, 2018, Bertero et al, 2019) highlighted the importance of SLC1A3-mediated aspartate uptake in promoting tumor growth, indicating its clinical importance. Our manuscript

identified SLC1A3 in ASNase resistance and metastasis, which further underscores the role of SLC1A3 in cancer.

4. The authors need to test the cancer cells for glutamate dependency following perturbations in SLC1A3 as it is also the transporter for glutamate. Glutamate can enter the cells via other transporters and can be converted to aspartate and rescue the cells. Figure 1E as referred in the text (line 134) does not show how glutamate depletion affects proliferation in SLC KO treated with asparaginase.

Reply: We thank the reviewer for this important comment. We included data on glutamate dependency to the revised manuscript.

To address this request, we added to Fig 1G measurements of intracellular glutamate levels under different conditions (3rd panel). We show that loss of SLC1A3 causes significant reduction in glutamate level in PC3 cells, confirming SLC1A3 function as glutamate transporter. Moreover, following ASNase treatment we also observed, similar to aspartate, reduced glutamate levels (Figure 1G). Lastly, the combination of ASNase treatment with SLC1A3 inhibition in cells expressing SLC1A3 potently caused a reduction of both intracellular aspartate and glutamate levels (Figure 3D). Altogether, these results indicate the induction of aspartate and glutamate dependency by ASNase. We incorporated these results in the text throughout the revised manuscript.

There is also no measurement of glutamate uptake, which is critical to evaluate in the SLC KO with and without asparaginase treatment. Also, did they try to do a rescue with esterified glutamate? They discuss it in the discussion (338) but do not show this result.

Reply: We thank the reviewer for this comment. To comply with this request, we first added the results from glutamate uptake measurements in control and SLC1A3-knockout cells to the revised manuscript (Fig 1F). It showed that SLC1A3 in PC3 cells is required for efficient glutamate uptake. Second, indeed the addition of esterified glutamate rescues the toxic effects of ASNase treatment applied to SLC1A3-knockout PC3 cells as well as to DU145 cells (which lack endogenous SLC1A3 expression) (Fig 1H and Fig 2I).

Specific comments:
Figure 1: what is sgNT? I understand it is a control but there is no de-abbreviation for it anywhere in the text. The legend of figure 1 does not match the panels.

Reply: We thank the reviewer for spotting these errors. Indeed, sgNT symbolizes the control sgNon-Targeting vector we used. Reference to it was now included in the revised manuscript. Also, we further corrected all issues with the text of the legends.

2D - no consistency in the correlation with the expression of SLC1A3 and sensitivity to asparaginase. In general, it is hard to interpret the figure because for each panel a different concentration of asparaginase was used.

Reply: We thank the reviewer for this comment. We observed a general trend where relatively high SLC1A3 mRNA levels indicated high basal aspartate uptake capability (Figs 2A–B). The exceptions in our cohort were LNCaP, SUM159PT and BT549 cells, with low SLC1A3 mRNA level but high basal aspartate uptake capacity. This can be explained by the relatively high expression of other aspartate/glutamate transporters in these cells (Fig 2C). Accordingly, SLC1A3-KO reduced aspartate uptake level only in SLC1A3-expressing cancer cells (Figs 2A–B). The sensitivity profiles of the tested cancer cell lines to ASNase treatment were generally consistent with the impact of SLC1A3-KO on aspartate uptake, with the exception of BT549 cells (Figs 2B and 2D).

Indeed, the various cell lines we used were treated with different ASNase concentration. As the referee acknowledged too, it is difficult to evaluate comparisons among different cancer cell lines as each displays a different sensitivity to ASNase. For Fig 2D, we first tested the sensitivity of each

cell line to ASNase, and then used a concentration that is close to toxic effects or has small effect on cell proliferation. In this panel, we were interested in the role of SLC1A3 and therefore mainly relied on the comparison between control (sgNon-targeting) and SLC1A3-KO and then how this would impact on the intrinsic ASNase effect within one cell line.

Figure 3- The authors need to show specificity of TFB-TBOA for SLC1A3. Broadly, it could have multiple targets. Specifically, it could be inhibiting aspartate SLC25A13 causing depletion in mitochondrial aspartate transport to the cytosol. This could also explain the depletion in NADH since this transporter participates in the malate aspartate shuttle.

B-C- It is hard to evaluate the comparison between the cells since the authors show cell cycle analysis for one cell line - PC3 and apoptosis analysis for the other two. Both assays should be shown for all 3 cell lines, preferably using the same method- FACS analysis that includes an antibody for apoptosis as PARP.

Reply: We appreciate the comments of the referee. First, we used the IncuCyte Caspase-3/7 apoptosis assay reagent (catalogue number: Essen 4440) to measure apoptosis in all three cell lines (Figure EV3B and EV3E, apoptotic cells were indicated in pink). This agent couples the activated caspase-3/7 recognition motif (DEVD) to NucView488, a DNA intercalating dye to enable quantification of apoptosis over time. We find this assay very reliable and accessible, as it is non-perturbing to cell growth and morphology. Kinetic activation of caspase-3/7 could be monitored and quantified using the IncuCyte basic analyzer and in parallel to the proliferation assays.

For the second point, we already discussed in the first manuscript version that the combination of ASNase with SLC1A3 inhibition can induce cell cycle arrest or apoptosis, depending on intrinsic characteristics of each cell line.

Also, I would expect nucleotide shortage to induce cell cycle arrest in G1 and not S. Please explain the result.

Reply: We appreciate the comment of the referee. The cell cycle distribution experiments were done with PC3 cells. These cells lack functional p53, and consequently, may have other perturbations that could prohibit prolonged arrest in G1, leading to an arrest in S phase. On top of it, the adverse perturbations by combinational SLC1A3 inhibition and ASNase were not restrictive to nucleotide shortage. Endogenous levels of asparagine, glutamine, aspartate and glutamate were all depleted. Consequently, perturbations in metabolites involved in urea cycle, TCA cycle, oxidation, glycolysis and carnitines metabolisms were observed (Figures 3E and EV4A-G). This suggested that nucleotides synthesis might only be one of the vulnerabilities caused by combinational treatment. In line with this, our attempts to restore cell proliferation with pyrimidine precursors failed in PC3.

3E- In addition, the authors need to show that the urea cycle enzymes are expressed in the cancer cells they are using as these are often not expressed in tumours.

Reply: We thank the reviewer for this comment. We examine the expression of all five key enzymes in the urea cycle: carbamoyl-phosphate synthetase 1 (CPS1), ornithine transcarbamylase (OTC), argininosuccinate synthetase (ASS1), argininosuccinate lyase (ASL), and arginase 1 (ARG1). Of these enzymes, ASS and ASL were expressed in all the cell lines we used for metabolic measurements (as determined by mRNA-seq data, see below). Additional data from the flux assay (see figure below) also indicated active incorporation of carbon and nitrogen for urea cycle, indicating the activity of the urea cycle enzymes (A: aspartate; G: glutamate; L-A: labeled aspartate; L-G: labeled glutamate).

Main urea cycle enzymes expression levels in PC3 cells under different conditions

gene	PC3 Ctrl	PC3+ASNase	PC3+TFB-TBOA	PC3+ASNase+TFB-TBOA
CPS1	1119,22	1125,07	1007,03	743,79
OTC	0	0	0	0
ASS1	6558,21	7219,3	6156,3	8060,68
ASL	380,8	378,82	356,78	416,95

ARG1	0	0	0	0
------	---	---	---	---

Main urea cycle enzymes expression levels in DU145 cells under different conditions

gene	DU145 mock	DU145+ASNase	DU145+TFB-TBOA	DU145+ASNase+TFB-TBOA
CPS1	1,84	8,8	1,73	7,11
OTC	0	0	0	0
ASS1	508,23	557,86	468,77	597,02
ASL	627,21	491,27	581,86	439,64
ARG1	0,92	0	0,86	1,02

Main urea cycle enzymes expression levels in DU145-SLC1A3oe cells under different conditions

gene	DU145-SLC1A3oe mock	DU145-SLC1A3oe+ASNase	DU145-SLC1A3oe+TFB-TBOA	DU145-SLC1A3oe+ASNase+TFB-TBOA
CPS1	6,82	4,46	5,04	10,03
OTC	0	0,89	0	0
ASS1	538,4	458,19	544,49	465,11
ASL	699,7	470,67	442,65	272,68
ARG1	0	1,78	3,02	0,91

Flux assay also indicated active incorporation of carbon and nitrogen for urea cycle, indicating the expression of relative enzymes.

EV3- All staining needs to be quantified.

Reply: The quantification of EV3 staining was shown in Figure 3A–C.

Figure 4: The authors need to perform tracing with glutamine because it is the major TCA replenishing metabolite that also contributes to urea cycle intermediates and pyrimidine synthesis.

Reply: We appreciate the comments the referee has made. We supposed the referee suggested to trace glutamate. Indeed, we also traced glutamate as well as aspartate (results are included in Figure EV5A–C). Except for some perturbations in glutamine, in general, ASNase treatment did not influence glutamate usage. We added text to the manuscript to address this issue.

Figure 5- It is unclear which perturbation is compared to which in terms of pathway analysis. Is there an explanation the authors have as to why A and C with double treatment look opposite? I would expect them to share mechanisms since they both express slc1a3.

Reply: We thank the reviewer for this comment. Indeed, the mechanisms among all three cell lines are similar, and the problem was color indications. In the revised manuscript we uniformed the color order for Figure 4A according to Figure 4B-C.

Also, upregulation of p53 as is seen in A should support a G1 or G0 arrest and not a prolonged S phase as is shown in figure 3.

Reply: We appreciate the comments the referee has made. Indeed, the gene signature of p53 effectors was upregulated in PC3 cells under combinational treatment. Those p53 effectors (from RNA-seq data) included p21 (cyclin dependent kinase inhibitor 1A), DDIT4, GDF15, GADD45A, TP63, ATF3 and SCN3B, which were further validated by RT-qPCR (Figure EV6A).

However, p53 was mutated in PC3 cells, and therefore we suspect that p63 (a family member to p53) might influence cell cycle progression. As discussed above, this combinational treatment causes a myriad of metabolic changes, making it hard to predict which phenotype will prevail and induce the arrest.

Figure 6- 4T1 can be injected to immune competent mice (BALB/c). It would make the data more relevant since the results will factor in the role of the immune system in asparaginase efficiency. For consistency, they can use 4T1 with luciferase to evaluate metastasis formation in a mouse with functional immune system.

Reply: Recent study reported that macrophage-mediated clearance of ASNase *in vivo*, indicating that immune system might influence ASNase effectivity (van der Meer et al, 2017). We therefore reason that the role of the immune system in ASNase efficiency is beyond the scope of this manuscript.

Referee #2:

Summary: Sun et al. characterize potential resistance mechanisms to L-asparaginase treatment in solid tumors utilizing a Crispr/Cas9 based approach in the PC3 prostate cancer cell line. Their genetic screen highlights the expression the glutamate/aspartate transporter SLC1a3 as a possible resistance mechanism. The authors show that genetic or pharmacologic inhibition of SLC1a3 increases sensitivity to L-asparaginase treatment in several cell lines and over-expression can confer resistance. Additionally, the authors show that SLC1a3 expression in breast cancer cells promotes tumor growth and progression alone and when treated with L-asparaginase.

The basic findings of this study are compelling and important to the field of cancer biology and cancer metabolism, however this work needs further development and requires significant revisions.

We thank the reviewer for the encouraging comments and the appreciation of our study.

Revisions:

1. In Figure 1E it appears that all of the cell line variants (control, sgSLC1a3, ASNase treated, sgSCL1a3/ASNase treated) proliferate at roughly equivalent rates up until day 3 and then diverge, with ASNase treated and sgSLC1a3/ASNase treated cells proliferating more slowly. Asparagine and aspartate levels in these conditions are shown after 3 days of treatment with sgSLC1a3 and ASNase treatment decreasing both as expected (Figure 1G).
a. Does it take 3 days for cells treated with ASNase and sgSLC1a3/ASNase 3 days to deplete intracellular asparagine/aspartate stores, which slows proliferation?

Reply: We thank the reviewer for this insightful comment. To answer this question, we seeded control and SLC1A3-KO PC3 cells in the absence and presence of ASNase and harvested at different time points post treatment (2hrs, 4hrs, 8hrs, 12hrs, 24hrs and 48hrs) for liquid-chromatography mass spectrometry (LC-MS). In order to collect cells at early time points (attached to the plate), the cells were seeded 24hours before the treatments were started. The results below demonstrate that asparagine was already significantly depleted after 2 hours incubation with ASNase. This was due to the enzymatic property of ASNase. In contrast, the depletion of aspartate was milder and more progressive along the incubation time, indicating the indirect stimulation in aspartate consumption following ASNase treatment.

Or do control and sgSLC1a3 cells upregulate ASNS expression after 3 days and increase biosynthesis of asparagine to boost proliferation rates in comparison?

Reply: We thank the reviewer for this comment. From RNA-seq analysis, ASNS induction by ASNase is similar (1.6- and 1.9- fold, respectively) in control and SLC1A3 inhibited PC3 cells, and therefore cannot explain the difference in proliferation observed after this time point.

b. What do intracellular levels of asparagine and aspartate look like before and after 72hrs?

Reply: The intracellular levels of asparagine and aspartate before 72hrs were shown above. Below we show the intracellular asparagine and aspartate level in control and SLC1A3-KO PC3 cells in the absence and presence of ASNase for 96hrs. Aspartate depletion is now much stronger than before.

2. Over all the authors only focus on SLC1a3's ability to transport aspartate as the primary mechanism by which SLC1a3 expression confers resistance to ASNase treatment. However, SLC1a3 also transports glutamate, which is a necessary nitrogen donor to synthesize asparagine from aspartate. It is likely that intracellular glutamate levels also play a large role in dictating resistance or sensitivity to ASNase treatment.

Reply: We agree with the reviewer's comments, and incorporated data on glutamate throughout the manuscript.

a. Figure 1F - How is glutamate uptake effected?

Reply: We included a panel for glutamate uptake to Figure 1F in the revised manuscript. The results showed that SLC1A3 loss-of-function resulted in decreased glutamate import.

b. Figure 1G - what do glutamate levels look like in these cell lines and after treatment with ASNS?

Reply: We included the data for glutamate to Figure 1G and discussed the results in the revised manuscript.

c. Figure 1H - Does glutamate supplementation also rescue sensitivity to ASNase treatment? As aspartate supplementation doesn't fully rescue proliferation defects after ASNase treatment it is likely that increasing glutamate levels would provide an additional growth benefit. Does glutamate + aspartate supplementation fully rescue ASNase treatment?

Reply: We agree with this comment of the reviewer and added the data to Figure 1H and discussed the results in the revised manuscript. Figure 1H show that supplementation of esterified glutamate alone (at the same concentration as aspartate) could fully rescue the ASNase-induced proliferation defects, indicating glutamate + aspartate certainly could fully rescue ASNase treatment. The somewhat higher efficiency of esterified glutamate compared to esterified aspartate could have multiple reasons. Technically, it can be that esterified glutamate is more readily used by the cells than esterified aspartate; or biologically, ASNase could cause a greater reduction in aspartate than glutamate (Figure 1G), and glutamate and aspartate are interconverted.

d. Figure 2J - Does glutamate alone or in combination with aspartate rescue growth after ASNase treatment in DU145 cell line?

Reply: We added the data for glutamate supplementation in DU145 cells to Figure 2I in the revised manuscript. Esterified glutamate alone could restore cell proliferation under ASNase treatment. Moreover, aspartate + glutamate could also rescue the adverse effect of ASNase in DU145 cells (the results were shown below).

e. Figure 3D - how does the SLC1a3 inhibitor TFB-TBOA effect glutamate levels?

Reply: The effect of TFB-TBOA alone on glutamate level was very mild. This probably was due to the short drug exposure time compared with the genetic knockout of SLC1A3. However, TFB-TBOA addition to PC3 cells under ASNase treatment further depleted intracellular aspartate and glutamate levels as expected. We added these results to Figure 3D and discuss it in the revised manuscript.

3. Figure 1F - the western blot should be improved or omitted from the figures as it is not does not show convincing evidence of strong over expression of SLC1a3 in these cells.

Reply: We agreed with the reviewer and omitted the western blot result.

4. The manner in which the data is represented in Figure 4 is overly complicated for a main figure and difficult to interpret even if the reader has knowledge/experience with this type of experiment and analysis.

a. Stacked graphs should be moved to supplemental figures and percent enrichment for the most relevant isotopomer for each metabolite should be shown for carbon and nitrogen individually.

Reply: We appreciate the comments the referee has made to help us refine the manuscript and adjusted the presentation of the tracing experiments and included the results as Figure EV5A-C in the revised manuscript.

b. The DU145 cell line used in this analysis is cultured in RPMI, however for the labeling experiment the authors use 1.5mM C13/N15 glutamine and 1.5mM C13/N15 aspartate which are not consistent with the concentrations of these amino acids found in RPMI. Why are different concentrations chosen for this experiment? Using different concentrations of these two metabolites for this experiment makes this data not directly relatable to the other experiments with this cell line throughout the paper.

Reply: We appreciate the comments the referee has made. Optimization experiments in DU145 cells using esterified amino acids showed that the addition of aspartate and glutamate at 1.5mM, but not 0.75mM, was sufficient to almost fully overcome ASNase inhibitory effect (shown below). This is why we chose these concentrations for the tracing experiments.

c. Figure 4a shows that cells secrete a significant amount of aspartate, which increased by ASNase treatment. Can the authors comment on why this is occurring?

Reply: The increase in unlabelled aspartate can be explained by the enzymatic activity of ASNase converting asparagine in the medium to aspartate.

d. This figure shows that extracellular aspartate does not contribute to the asparagine pool. If cells are exclusively relying on endogenously produced sources of asparagine why does ASNase treatment have an effect? And why does SLC1a3 inhibition increase this effect if its primary function is to import aspartate?

Reply: We thank the reviewer for this comment. ASNase could deplete exogenous and endogenous asparagine (Figure 1G and please refer to reviewer#3 major point#1). Acute lymphoblastic leukemia (ALL) cells are auxotrophic for asparagine and therefore are sensitive to asparagine depletion by ASNase. However, in PC3 control cells, despite of asparagine depletion (Figure 1G), cells remained in a proliferative state (resistant). By performing a genome-wide functional screen, we identified SLC1A3-mediated fueling of aspartate/glutamate as a key contributor to this phenotype. Impact on the urea cycle, nucleotide synthesis, TCA cycle, oxidation, glycolysis and carnitine metabolisms were documented (Figure 3E). In line with this, we observed increased consumption of aspartate/glutamate following ASNase treatment, indicating high dependency on exogenous aspartate/glutamate import by SLC1A3.

There is no difference in labeling between control and ASNase treated cells in any of the other detected metabolites again begging the question of how aspartate supplement is boosting proliferation of cells treated with ASNase. This further points to the fact that glutamate uptake mediated by SLC1a3 may play an important role in this system.

Reply: We completely agree with this comment from the reviewer. Indeed, glutamate supplementation could also restore cell proliferation in SLC1A3-KO PC3 and DU145 cells following ASNase treatment (as aspartate supplementation), highlighting the importance of glutamate in ASNase resistance. However, except for some perturbations in glutamine incorporation, the profiles of isotopologues derived from labeled glutamate, in general, remained the same between mock and ASNase conditions. The results were incorporated into Figure EV5C and further discussed in the revised manuscript.

5. There is an additional supplemental figure, EV6 that is not referenced in the text. All references to Figure EV7 are incorrect in the text and mistakenly refer to EV6.

Reply: We apologize to the reviewer for this mistake, and corrected it in the revised manuscript.

6. Figure 6a - The Sum159PT cell line is only used to show asparagine levels in the serum, fat pad and tumor. Does this cell line respond to ASNase treatment in vivo?

Reply: We thanks the reviewer for this comment. We used SUM159PT cells in a pilot experiment to test whether ASNase could deplete asparagine and glutamine levels within the growing tumors and the tumor growing environment. The ASNase treatment lasted for 5 consecutive days when the tumor volume reached~ 250mm³, and then the mice were sacrificed and samples were collected and subjected to liquid-chromatography mass spectrometry (LC-MS).

Figure 2C in the revised manuscript shows that SUM159PT cells do not depend on SLC1A3 expression for aspartate and glutamate transportation, probably due to the high expression of other SLC1A3 homologues: SLC1A1 and SLC1A7. Follow up experiment will be needed to address whether these homologues are involved in ASNase resistance in the future.

This data would be better if produced from animals with 4T1 tumors as this cell line is used to generate the remainder of the in vivo functional data.

Reply: We indeed collected 4T1 tumors and measured intra-tumor amino acids by liquid-chromatography mass spectrometry (LC-MS). The results are shown in Figure 5C of the revised manuscript.

7. Figure 6b - only a single time point is shown for response to ASNase treatment. How do these tumors respond over time and what do the growth kinetics look like with ASNase treatment.

Reply: We appreciate the comment of the reviewer. The tumor volumes were measured every 3 days. The data from day 9 and day 12 were included in the previous manuscript at Figure6B (now

Figure 5B) and Figure EV6B (Figure EV7B). In these experiments we found a clear response to ASNase only at early stage of tumor growth. Later on, there is a trend but it is not significant anymore. We provide the full data to the reviewer below.

a. The authors performed primary tumor resection and survival studies using the 4T1 model which readily metastasizes. Did the authors collect lungs from these animals to perform histological analysis of metastatic lesions? This would be excellent complimentary data to the experimental metastasis model with MDA-MB231 and would show ASNase effect on spontaneous metastasis.

Reply: We appreciate the comment of the reviewer. This would have been a good experiment if the mice were sacrificed at the same time. However, the 4T1 experiment was aimed at survival and animals were sacrificed only when showing breathing problems. The analysis of the lungs showed lots of metastasis in all conditions.

8. Overall the figures need more descriptive labels so that it is easy for the reader to understand what cell lines, treatments, or conditions are being represented without having to refer the text or the figure legends

Reply: We appreciate the comment of the reviewer. In the revised version of the manuscript, we paid extra attention to the presentation of the figures and text.

Referee #3:

In this manuscript by Sun et al., the authors describe a novel role for SLC1A3-mediated aspartate uptake in resistance of solid tumors to L-asparaginase treatment. By performing a whole genome CRISPR-Cas9 screen in a prostate cancer cell line, the authors find that one of the genes whose knockout sensitizes cells to ASNase treatment is plasma membrane glutamate/aspartate transporter SLC1A3. After validating that the increase in intracellular aspartate levels mediated by expression of this transporter is responsible of ASNase resistance in many different cells by genetics and pharmacologic means, the authors engage in showing the relevance of this resistance in vivo. Knockout of SLC1A3 in prostate cancer cells impairs tumor growth upon ASNase treatment, and concomitant overexpression of the transporter in a breast cancer cell line that do not express basal SLC1A3 enables these tumors resistant to the treatment both in xenograft and orthotopic models. Intriguingly, the authors find that overexpression of SLC1A3 strongly increases metastatic burden in the same breast cancer model, in agreement with recent studies showing that ASNase treatment and asparagine availability have an impact in metastasis. Of note, the authors provide a lot of useful data, such as metabolic changes or transcriptional responses upon ASNase treatment; and validate their results in different cancer types.

Major points:

1. ASNase is known to break down not only asparagine but also glutamine at certain concentrations. This is a critical point not addressed throughout the manuscript, since SLC1A3-mediated uptake of glutamate would also protects against glutamine depletion. Because of this, repeating some experiments and measuring some extra metabolites is recommended. A) Treat PC3 cells in vitro with the L-ASNase concentration used in your first assays (0.3-0.5 U/mL) and measure not only depletion of asparagine but also glutamine in the media.

Reply: We agree with this comment of the reviewer. Indeed, both endogenous asparagine and glutamine levels were robustly depleted following ASNase treatment in PC3 cells (Figure 1G in the

revised manuscript). Moreover, from the same flux assays (Figure EV5), we observed that ASNase could deplete asparagine and glutamine both intracellularly and in the medium. We provide the figure to the reviewer below.

B) Validate that SLC1A3 knockouts sensitize cells to asparagine depletion by using an ASN-depleted media instead of ASNase treatment. ASN depletion by this means should phenocopy ASNase treatment effect bypassing any potential depletion of Glutamine.

Reply: Figure 1G shows that SLC1A3-KO depletes intracellular glutamine. While we do not know the exact reason for this phenomenon (possibly due to glutamate shortage for glutamine synthesis), this fact discards the single role of glutamine as these cells proliferate as control cells in the absence of ASNase. This glutamine shortage by SLC1A3 knockout also prevents discrimination in asparagine or glutamine depletion by ASNase.

C) It is striking that SLC1A3 KO shows such a decrease in intracellular glutamine levels (Fig. EV1C), raising the possibility that SLC1A3 rescue of ASNase treatment could be more related to glutamine depletion than to asparagine, at least in the model. Could the authors repeat these experiments of proliferation in a media with asparagine and aspartate, but not with glutamate in it? Given the almost identical structure of glutamate and aspartate, glutamate (first product after glutaminase reaction) can be taken up by SLC1A3 and would definitely rescue GLN-deprivation phenotype. Conversely, do the same experiments in the presence of asparagine and glutamate, but not aspartate, in the media.

Reply: We thank the reviewer for this comment.

Indeed, SLC1A3-KOs significantly reduced intracellular glutamine levels (Figure 1G). However, this did not affect cell proliferation of these cells, suggesting the importance of asparagine availability in this model. This is consistent with a previous study, where asparagine became an essential amino acid when glutamine was deleted (Pavlova et al, 2018). In addition, in the revised manuscript, we included results that indicate that supplementation of either cell-permeable aspartate or glutamate could restore cell proliferation of ASNase-treated SLC1A3-KO PC3 cells as well as of DU145 cells (results added to Figure 1H and Figure 2I). Lastly, the ectopic expression of SLC1A3 rescued the toxic effect of ASNase on DU145 cells, and as the referee proposed, promoted glutamine synthesis. However, asparagine and glutamine were still depleted in DU145-V5-SLC1A3 cells, probably due to the efficient enzymatic property of ASNase (Figure EV4A).

D) In the in vivo experiments shown in Fig. 6 (Fig. 6A and 6C) with analysis of orthotopic tumors metabolite levels, could the author provide the levels of Glutamine and Glutamate?. This would help to see if Gln is depleted in the tumor and if SLC1A3 may rescue by taking up glutamate.

Reply: We thank the reviewer for this comment. For Figure 6A (Figure 5A in the revised manuscript), SUM159PT cells were orthotopically injected into the mice mammary fat pad and when the tumor reached $\sim 250\text{mm}^3$, ASNase treatment was conducted for 5 consecutive days (3 mice per group). Below we provide the data for glutamine and glutamate from the SUM159PT tumor samples (glutamine results were also included in Figure EV7A). Even though there is a slight reduction in glutamine and glutamate levels following ASNase treatment, this is not significant (the p value was calculated by two-tailed unpaired t test in Prism 7). This might be due to the replenishment of glutamine by many aspartate/glutamate transporters present in SUM159PT cells: SLC1A1, SLC1A3 and SLC1A7 (Figure 2C).

For Figure 6C (Figure 5C in the revised manuscript), mice were pretreated with ASNase for 2 days before the cancer cells were injected into the mice mammary fat pad, and ASNase treatment was continued till the mice were sacrificed (n=13 for each group except for 4T1+ASNase, n=12). Here we observed a clear and significant effect of ASNase on both glutamine and glutamate depletion. Indeed, SLC1A3 expression significantly negated the glutamine reduction at the cost of glutamate usage as the reviewer suggested (results are based on 5 xenograft tumor samples and p-values were calculated by two-tailed unpaired t-test in Prism 7).

Based on above results, we reason that SLC1A3 imports aspartate and glutamate, and promoted ASNase resistance by directly negating intracellular aspartate depletion and indirectly replenishing intracellular glutamine. We added the results to Figure 5C and discussed this point in revised manuscript.

E) The authors mention in the discussion that addition of esterified Glutamate also rescues ASNase treatment. Could the authors show these results adding it to Fig. 1H and discuss the possibilities further?

Reply: We agree with the reviewer and added this information to our manuscript (Figure 1H). Please see our response to Reviewer #1 comment #4.

2. Could the authors comment why they think SLC25A1 scores? This is a mitochondrial citrate carrier that due to its function is required for reductive carboxylation (Jiang et al., 2016, Nature; PMID: 27049945). Is glutamine metabolism rewired from oxidative to reductive carboxylation during ASNase treatment?

Reply: We thank the reviewer for this comment.

To our limited knowledge, we could only speculate at this point that either stimulated consumption of aspartate/glutamate induced by ASNase promoted the reductive carboxylation for aspartate synthesis, or that SLC25A,1 as a mitochondrial citrate-malate exchanger, is key for TCA cycle under ASNase conditions in PC3 cells.

3. In Figure 3E, the authors stress that NADH levels decrease after ASNase treatment, and that this is indicative of a defective aspartate synthesis/metabolism. Even though this result would make sense, it is hard to believe that NADH levels decrease that much in the absence of appreciable changes in NAD⁺ levels after such a short treatment. NADH should be converted into NAD⁺ and the results shown in the figure suggest that the total pools change. The authors should show the results as NAD⁺/NADH ratio as previously done in similar studies, and probably repeat it in the presence of ASNase or the double treatment but using a fluorescent detection kit instead of mass spectrometry.

Reply: We thank the referee for this comment. When PC3 cells were treated with ASNase and SLC1A3 was inhibited (Figure 3E), the perturbations in glycolysis indicated depleted glyceraldehyde3P levels, which are involved in lactate synthesis (schemes included in Figure 3E). Due to the lack of glyceraldehyde3P, the synthesis of NADH from NAD⁺ is likely to be impaired, which can explain why NAD⁺ was not consumed. There was a very mild increase of NAD⁺ levels from the LC-MS analysis but not significant (Figure 3E).

We took the advice of the reviewer and performed the analysis of NAD⁺/NADH using a commercially available Enzychrom NAD⁺/NADH assay kit (E2ND-100). Following the protocol, around 1X10⁵ cells were pelleted and washed with cold PBS. However, even after incubation for 30 mins (15 mins as the standard incubation time according to the protocol), the measurements of OD565 were close to the background value (BG value: 0.05, samples between 0.06-0.08) which precluded a robust result. We conclude that liquid-chromatography mass spectrometry (LC-MS) is a more robust method for this experiment. And we calculated the NAD⁺/NADH ratio and included in the Figure 3E.

4. The results shown in Figure 4 are intriguing. imported isotope-labeled aspartate barely labels any metabolite at appreciable levels (including pyrimidines), but still rescues ASNase sensitivity without replenishing the asparagine pool. This result suggests that, at least in the solid tumors studied here, ASNase treatment does not affect tumor growth by depleting asparagine, but rather by depleting aspartate as a result of an increased demand of asparagine synthesis from aspartate. If that's true, supplying downstream metabolic outputs of aspartate, and not asparagine, should rescue sensitivity to ASNase. Could the authors for example supplement high levels of nucleosides (both purine and pyrimidine precursors, for which aspartate is required) and rescue ASNase treatment or ASN-depletion from the media? This should work at least in PC3 cells where an impaired cell cycle is observed, which could be due to a shortage in nucleotide precursors.

Reply: We thank the reviewer for this suggestion. We indeed supplemented the precursors for purine and pyrimidine synthesis (dC, dG, dA, and dT) and examined the effect of ASNase on cell toxicity. We used different combinations and concentrations but never observed a clear rescue. This result may fit well with the broad effect of ASNase on many cellular processes beyond nucleotide production. For example, ASNase treatment with SLC1A3 inhibition induced marked reduction in intra-cellular levels of asparagine, glutamine, aspartate and glutamate which affected metabolites involved in the urea cycle, TCA cycle, oxidation, glycolysis and carnitines metabolic pathways (Figure 3E and Figure EV4A-G).

Minor

points:

1. In line 208-209, the authors refer the text to Fig. 4A-G but it should be Fig. EV4A-D.

Reply: Indeed, there was "EV" missing. The Figure 4A-G has been replaced by Figure EV4A-G.

2. EV6 is not referenced. EV7 is referenced as EV6 in line 260.

Reply: The referred figures were adjusted.

3rd Editorial Decision

9th Aug 2019

Thank you for submitting the revised version of your manuscript. My sincere apologies again for the delay in processing your revised manuscript due to protracted referee input. Your revised study has been re-evaluated by two of the original referees, whose comments are enclosed below. As mentioned, the third referee was not able to send his-her report at this time. Please note however, that we have assessed your response to his-her concerns editorially and found these to be reasonably addressed.

As you will see the referees find that their criticism has been sufficiently addressed and they are now in favor of publication. Overall, we are thus pleased to inform you that your manuscript has been accepted in principle for publication in The EMBO Journal, pending minor revision.

Where referee #1 remains more hesitant regarding the translational claims of your study, we have re-considered the matter here in the team and concluded this can be settled satisfactorily in a minor revision by complementary discussion of the findings and introducing caveats where appropriate.

 REFEREE REPORTS:

Referee #1:

Revised manuscript

The authors corrected the discrepancy between the referral figure numbers in the text and the actual figure numbers.

Remaining issues:

1. In response to my previous comment, that authors analysed the TCGA to show high SLC expression in resistant tumors and low expression in ALL. They succeeded in showing low expression in ALL yet found high expression in renal, liver and stomach cancers and not in prostate and breast which is the cancers used in their paper. They claim their findings are more relevant to metastasis, so why did they not look at the TCGA dataset if cancers metastasis express high SLC? This makes the translational relevance of their findings in cancer cells questionable in regards to actual tumors.
2. The authors now show results for the glutamate depletion following loss of SLC. This may suggest that all their findings regarding aspartate could be secondary to the deficiency in glutamate since aspartate can be synthesized from glutamate. I would phrase their conclusions cautiously.
3. For some reason the authors show urea cycle intermediated but do not show arginine which is the main product of UC enzymes outside the liver.
4. The effect of TFB-TBOA on other aspartate and glutamate transporters as slc25a13 has not been addressed.
5. The authors claim the immune system effect on asparaginase efficacy is beyond the scope of this paper. If so, it makes it harder to highlight the translational relevance of the paper.

Referee #2:

The authors have addressed most of the reviewers concerns.

3rd Revision - authors' response

13th Aug 2019

The authors performed the requested editorial changes.

Corresponding Author Name: Reuven Agami
 Journal Submitted to: EMBO Journal
 Manuscript Number: EMBOJ-2019-102147